# PRODH safeguards human naive pluripotency by limiting mitochondrial oxidative phosphorylation and reactive oxygen species production

Cheng Chen [1,2,8], Qianyu Liu [3,8], Wenjie Chen [4,8], Zhiyuan Gong [5], Bo Kang [6], Meihua Sui [2✉], Liming Huang [1✉] & Ying-Jie Wang [6,7✉]

## Abstract

Naive human embryonic stem cells (hESCs) that resemble the pre-implantation epiblasts are fueled by a combination of aerobic glycolysis and oxidative phosphorylation, but their mitochondrial regulators are poorly understood. Here we report that, proline dehydrogenase (PRODH), a mitochondria-localized proline metabolism enzyme, is dramatically upregulated in naive hESCs compared to their primed counterparts. The upregulation of PRODH is induced by a reduction in c-Myc expression that is dependent on PD0325901, a MEK inhibitor routinely present in naive hESC culture media. PRODH knockdown in naive hESCs significantly promoted mitochondrial oxidative phosphorylation (mtOXPHOS) and reactive oxygen species (ROS) production that triggered autophagy, DNA damage, and apoptosis. Remarkably, MitoQ, a mitochondria-targeted antioxidant, effectively restored the pluripotency and proliferation of PRODH-knockdown naive hESCs, indicating that PRODH maintains naive pluripotency by preventing excessive ROS production. Concomitantly, PRODH knockdown significantly slowed down the proteolytic degradation of multiple key mitochondrial electron transport chain complex proteins. Thus, we revealed a crucial role of PRODH in limiting mtOXPHOS and ROS production, and thereby safeguarding naive pluripotency of hESCs.

**Keywords** Electron Transport Chain Complex; Human Naive Pluripotency; mtOXPHOS; PRODH; Reactive Oxygen Species
**Subject Categories** Metabolism; Stem Cells & Regenerative Medicine

## Introduction

Human embryonic stem cells (hESCs), derived from the inner cell mass (ICM) of the developing blastocyst, are characterized by their unlimited capacity for self-renewal as well as the potential to differentiate towards all three germ layers (Young, 2011). Due to dynamic changes in pluripotency during embryonic development, two distinct pluripotency states have been designated, namely naive and primed pluripotency (Nichols and Smith, 2009). Compared to conventional primed hESCs that resemble the post-implantation epiblasts, naive hESCs correspond to inner cell mass (ICM) of the pre-implantation blastocyst (Huang et al, 2014; Nichols and Smith, 2009; Stirparo et al, 2018a) and exhibit more remarkable plasticity and unbiased differentiation potential (Guo et al, 2021; Io et al, 2021; Lee et al, 2017; Yang et al, 2016), thus providing a valuable model for developmental studies and regenerative medicine. There are several critical features in naive hESCs relative to their primed counterparts: global DNA hypomethylation, the preferential activity of the OCT4 distal enhancer, two active X chromosomes without the H3K27me3 mark, and a bivalent metabolic system (Collier and Rugg-Gunn, 2018; Theunissen et al, 2016; Tsogtbaatar et al, 2020). Interestingly, despite the immature appearance of mitochondria in the naive state, naive hESCs rely on both oxidative phosphorylation (OXPHOS) and glycolysis, while primed hESCs are highly glycolytic (Tsogtbaatar et al, 2020; Zhang et al, 2018; Zhou et al, 2012). Recent findings have revealed the critical role of mitochondria in influencing the cell fate and naive-primed state interconversion of ESCs (Arnold et al, 2022; Bahat et al, 2018; Carbognin et al, 2016; Kim et al, 2022; Martinez-Val et al, 2021; Peron et al, 2021; Pezet et al, 2021; Zhou et al, 2012). Promoting glycolysis by ectopic HIF1 expression or exposure to hypoxia can speed the mESC transition to the primed state (Zhou et al, 2012). Indeed, STAT3 activation promotes the increased growth of mESCs and the reversion of primed stem cells back to a naive pluripotent state by upregulating mitochondrial genes and increasing mitochondrial oxidative metabolism, indicating a crucial function for

[1]Shaoxing People's Hospital, Shaoxing Hospital, Zhejiang University School of Medicine, Shaoxing, Zhejiang 312000, China. [2]School of Basic Medical Sciences, Zhejiang University, Hangzhou, Zhejiang 310058, China. [3]College of Life Sciences, Zhejiang University, Hangzhou, Zhejiang 310058, China. [4]Department of Obstetrics and Gynecology, Sir Run Run Shaw Hospital, School of Medicine, Zhejiang University, Hangzhou, Zhejiang 310016, China. [5]Department of Oral and Maxillofacial Surgery, The First Affiliated Hospital, Zhejiang University School of Medicine, Hangzhou 310003, China. [6]State Key Laboratory for Diagnosis and Treatment of Infectious Diseases, National Clinical Research Center for Infectious Diseases, Collaborative Innovation Center for Diagnosis and Treatment of Infectious Diseases, The First Affiliated Hospital, School of Medicine, Zhejiang University, Hangzhou, Zhejiang 310003, China. [7]Cancer Center, Zhejiang University, Hangzhou, Zhejiang 310058, China. [8]These authors contributed equally: Cheng Chen, Qianyu Liu, Wenjie Chen. ✉E-mail: suim@zju.edu.cn; 0622122@zju.edu.cn; yingjiewang@zju.edu.cn

mitochondrial metabolism in naive pluripotency maintenance (Carbognin et al, 2016). However, the mechanisms of mitochondrial action and its molecular regulators in naive hESCs remain largely unknown.

Proline dehydrogenase (PRODH) or proline oxidase (POX), an enzyme that co-localizes with complex II in mitochondrial inner membranes, catalyzes the first step in proline degradation to produce pyrroline-5-carboxylate (P5C) (Kramar, 1971). As P5C is an essential intermediator in the metabolic interconversions between the tricarboxylic acid (TCA) cycle and the urea cycle, the proline metabolic axis mediated by PRODH is crucial in regulating various biological functions (Adams, 1970; Phang, 1985). Meanwhile, PRODH can also transfer electrons from proline to Coenzyme Q1 (CoQ1) with a flavine adenine dinucleotide at the active site (Hancock et al, 2015; Phang, 2019). Then, electrons can continue to pass through the mitochondrial electron transport chain (ETC) to produce ATP or directly reduce dissolved oxygen at complex III to form reactive oxygen species (ROS) (Finkel, 2011; Hancock et al, 2015). ROS have a crucial role in cellular signaling when present at low levels; however, excessive accumulation of ROS can lead to irreversible oxidative damage to lipids, proteins, and DNA, interfering with vital cellular functions (Fruehauf and Meyskens, 2007). Therefore, it has been reported that PRODH plays important roles in apoptosis, senescence, and cancer treatment via cellular redox control (Huynh et al, 2022; Liu et al, 2006; Misiura et al, 2021; Nagano et al, 2017). However, the physiological functions of PRODH in stem cells, especially its effects on ESC pluripotency, are still largely unknown.

In the present study, we revealed that PRODH expression is upregulated in naive hESCs. Such upregulation contributes to maintaining the pluripotency and proliferative capacity of hESCs in the naive state over the primed state. Besides, PD0325901, an essential component in naive culture medium, upregulates the transcripts of PRODH by reducing c-Myc (herein termed as MYC). By comparing the proline metabolic pathway profiles between naive and primed hESCs, we ruled out the possibility that PRODH impacts naive pluripotency through its regulation of proline metabolism per se. Remarkably, mitochondrial oxidative phosphorylation (mtOXPHOS) is enhanced by down-regulating PRODH in naive hESCs, leading to elevated production of ROS. Excessive ROS can trigger a cascade of detrimental effects on naive hESCs, including autophagy, DNA damage, and apoptosis, impairing cells' pluripotency and hindering their ability to proliferate. Finally, we discovered the negative correlation between PRODH and complex I-IV in ETC, and elucidated the mechanism underlying the enhancement of mtOXPHOS upon PRODH knockdown (KD) in naive hESCs.

# Results

## PRODH specifically regulates pluripotency and proliferation of naive hESCs

First, we conducted a literature search to collect publicly available RNA-seq data (Stirparo et al, 2018b). Compared with the primed state, PRODH was expressed at a higher level in the naive state among different cell lines and culture media (Fig. 1A). For further

research, we cultured primed H9 hESCs in mTeSR1 and induced naive-like H9 by RSeT Feeder-Free Medium (designated as "Rset") or PXGL hESC induction system as described previously (Chen et al, 2022). Similarly, a significant upregulation in transcript and protein expression levels of PRODH was detected in Rset and PXGL versus mTeSR1 (Fig. 1B,C). To investigate the role of PRODH in maintaining naive pluripotency, we knocked down PRODH using shRNA (non-targeting shRNA control [PLKO.1] or shPRODH) in primed H9 and naive H9 (including Rset and PXGL) (Fig. 1D). Remarkably, PRODH KD significantly attenuated AP+ colony formation, reducing both colony size and number, exclusively in naive but not in primed cells (Figs. 1E and EV1A). Consistently, the fluorescence intensity of NANOG was significantly diminished in Rset-shPRODH cells and PXGL-shPRODH cells, while no such reduction was observed in primed shPRODH cells (Fig. 1F). Although the expression levels of OCT4 did not show significant differences between H9N-PLKO.1 and H9N-shPROSH, PRODH KD resulted in a substantial downregulation of NANOG, SOX2, and naive marker genes, suggesting impaired naive pluripotency (Fig. EV1B–D). Meanwhile, examination of germ layer markers revealed the differentiation of shPRODH-treated naive hESCs toward mesendoderm (Fig. EV1F). This differentiation trend was in line with the severe neurological phenotype exhibited by children carrying mutations in PRODH (Afenjar et al, 2007). In contrast, the total transcript and protein levels of OCT4, NANOG, and SOX2 in primed cells remained unchanged after PRODH KD (Fig. EV1B,E), indicating that PRODH does not affect the primed pluripotency.

With further research, we found that PRODH not only impacted pluripotency but also significantly influenced the proliferative capability of naive stem cells. Rset-shPRODH and PXGL-shPRODH cells showed limited proliferative capacity, marked by a notable decrease in the slope of the cell proliferation curve (Fig. 1G). Consistent with this result, the fluorescence of EDU staining was also decreased in Rset-shPRODH and PXGL-shPRODH cells, indicating a reduction in DNA replication activity (Fig. 1H and Appendix Fig. S1). In contrast, the primed hESCs had no such phenotype (Fig. 1G,H and Appendix Fig. S1).

Together, these results suggest that PRODH regulates the capacity for pluripotency and proliferation, specifically in naive but not primed hESCs.

## PD0325901 upregulates PRODH by reducing MYC in the naive state

To explore the cause of the elevated expression of PRODH in the naive state relative to the primed state, we set out to compare the key components in the three culture media. After reviewing the supplemented components in mTeSR1, Rset-feeder free, and PXGL culture systems, we identified four essential supplements shared between the Rset and PXGL systems but absent in mTeSR1: PD0325901 (MEK inhibitor), Gö 6983 (PKC inhibitor), Leukemia inhibitory factor (LIF), and N-2 Supplement (Fig. 2A). Upon treating primed hESCs with each of the four supplements at the indicated concentrations, the transcript and protein levels of PRODH were significantly upregulated only when PD0325901 (PD) was added to the mTeSR1 culture system (Fig. 2B,C). Similarly, compared to the other three supplements, PRODH was significantly decreased only in the PXGL system without PD

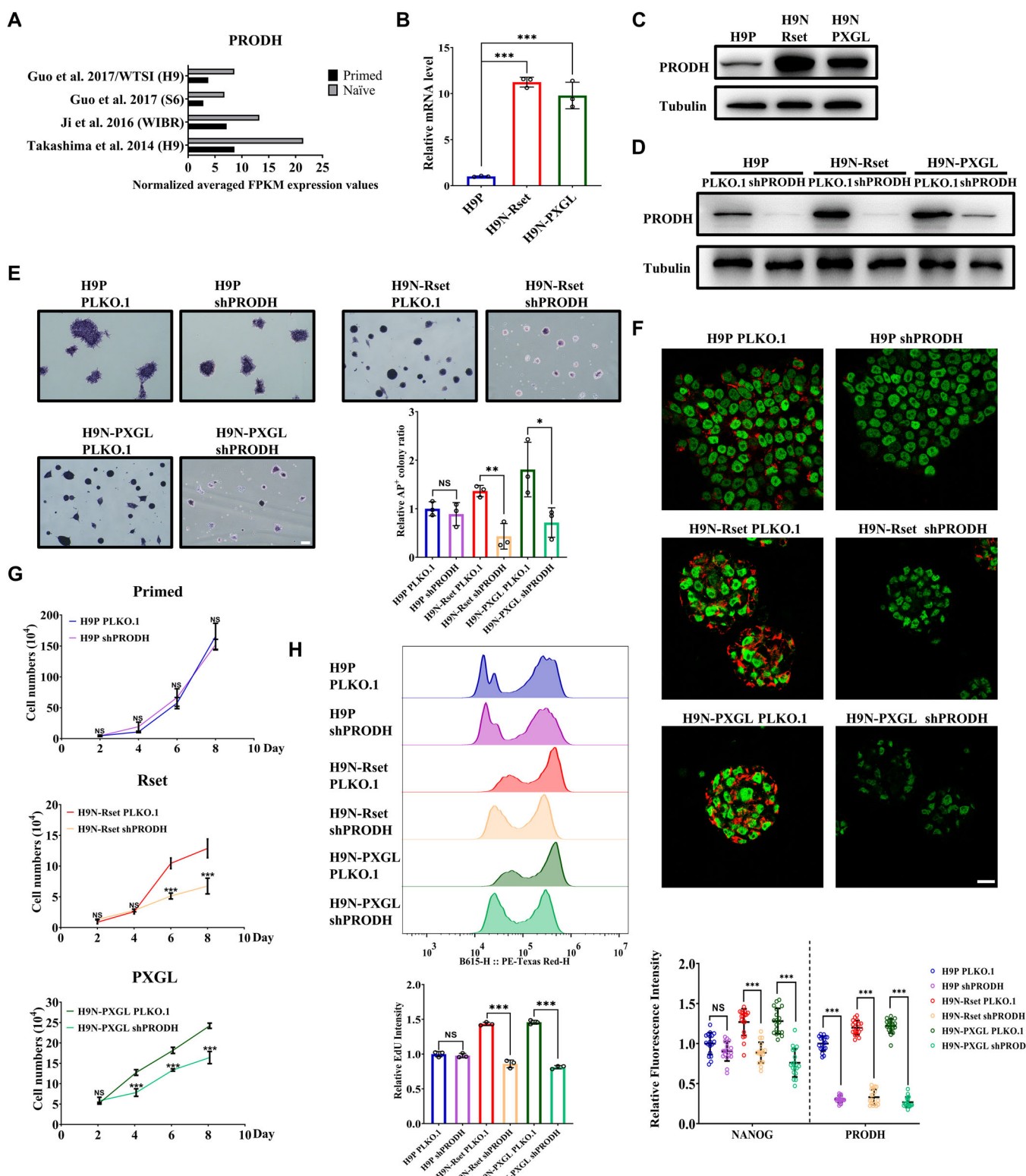

 Therefore, the addition of PD appeared to be a crucial factor for elevated PRODH levels in the naive culture system.

PRODH is known to be upregulated under stress conditions via various signals, such as P53 when DNA is damaged, AMPK when hypoxia and nutrient lack (Phang, 2019), and downregulated by MYC through miR-23b* (Liu et al, 2012). Next, we determined whether PD affected the expression of PRODH through the above pathways. PD reduced the protein level of MYC but did not affect

**Figure 1. PRODH specifically regulates human naive pluripotency.**

(A) Summary of RNA-seq data for PRODH in primed versus naive hESCs in the literature. (B) The mRNA levels of PRODH in primed and naive H9 hESCs were determined by qRT-PCR. (C) WB showing PRODH is overexpressed in naive H9 cells compared to primed cells. (D) WB showing PRODH KD efficiency by specific shRNAs transfected to primed (H9P) and naive (H9N) H9 cells cultured in Rset-feeder free or PXGL culture media. (E) Representative light micrographs of alkaline phosphatase positive (AP + ) colonies in PLKO.1 and shPRODH cells. Scale bars, 100 μm. Global variation in the number of AP+ colonies was presented in Fig. EV1A. The quantification of AP+ colonies in each condition was processed by ImageJ and normalized to H9P PLKO.1 values. (F) PLKO.1 and shPRODH H9 cells cultured in all three media were immunostained for NANOG (green) and PRODH (red). Representative confocal micrographs were shown. Scale bars, 20 μm. The experiment was conducted independently three times, and a total of 18 cell images from each condition were analyzed using ImageJ. The fluorescence intensity was quantified using the mean of the three independent experiments and normalized to H9P PLKO.1 values. (G) Cell proliferation curves of PLKO.1 versus shPRODH cells. $5 \times 10^4$ of primed and naive H9 hESCs were seeded in 12-well plates, and total cell numbers were counted every 48 h. (H) The cell proliferation rate represented by EdU incorporation was assessed using EdU assay. The fluorescent intensity of cells stained with EdU was determined by flow cytometry (upper), and the quantitative data was normalized to H9P PLKO.1 values (lower). Representative images of EdU stain were presented in Appendix Fig. S1A. Data information: The statistical significance was analyzed using unpaired two-tailed Student's t-test in panels (B, E, F, G, and H). NS, not significant ($P > 0.05$). *$P < 0.05$, **$P < 0.01$, ***$P < 0.001$. The data shown were from three independent biological replicates. In (B, E, G, and H), each data point represents an independent biological replicate. In (F), each data point represents the relative fluorescence intensity of cells derived from each experiment. Data were presented as mean ± SD. Source data are available online for this figure.

P53 and AMPK (Fig. 2F). Besides, both Rset and PXGL naive hESCs, as well as primed H9 cells treated with PD, showed a reduction in MYC mRNA levels, consistent with the high level of PRODH in these hESCs (Fig. 2G). Moreover, MYC KD enhanced transcript and protein expression levels of PRODH in primed H9 hESCs, just like PD treatment (Fig. 2H,I). It was previously shown in other cellular systems that MYC suppressed POX/PRODH expression primarily through upregulating miR-23b* (Liu et al, 2012), and the level of MYC is regulated by MEK/ERK pathway: ERK phosphorylates Ser62 and stabilizes MYC proteins (Farrell and Sears, 2014). Thus, there was compelling evidence to speculate that the MEK/ERK-MYC-PRODH signaling pathway may play a pivotal role in driving the difference in PRODH expression between primed and naive hESCs.

To test this hypothesis, we determined the PRODH levels in PD-treated H9P cells overexpressing ectopic MYC. While overexpressing MYC did not lead to a further reduction in PRODH, suggesting a potential limit of MYC's inhibitory effect on PRODH (H9P OE-MYC vs. H9P), it did, however, partially impede the elevation of PRODH induced by PD (H9P OE-MYC + PD vs. H9P + PD) (Fig. 2J). A summary of MEK/ERK-MYC-PRODH signaling pathway was presented in Appendix Fig. S2A. Taken together, we reveal here that PD0325901 critically upregulates PRODH level in the human naive state by suppressing MYC.

## Comparison of PRODH-mediated proline metabolism between naive and primed hESCs

Given PRODH's primary role in proline metabolism, we first examined the changes in amino acids and metabolic kinases in the proline metabolic pathway. In order to exclude the interference by proline content in different culture media, we chose Rset (L-proline concentration of 0.2 mM) and mTeSR1 (L-proline concentration of 0.216 mM) culture systems for further research. Targeted metabolomics of amino acids and their derivatives were analyzed (Fig. EV2 and Dataset EV1). In addition, proteomics (reported in our previous study (Chen et al, 2022)) and Western blot (WB) techniques were utilized to quantify and compare kinases associated with the proline metabolic pathway between naive and primed hESCs, including H1 and H9 cells. Surprisingly, despite the upregulation of PRODH and downregulation of PYCR1 in the naive hESCs (Fig. 3A,B), there was no significant difference in

proline levels between naive and primed states (Fig. 3C). Pyrroline-5-carboxylate (P5C), an obligate intermediate in the metabolic interconversions between the tricarboxylic acid (TCA) cycle and urea cycle, was only detected in naive hESCs (Fig. 3C). In the context of the urea cycle, a reduction in ornithine levels was observed in the naive state (Fig. 3C), aligning with the down-regulation of ornithine aminotransferase (OAT) (Fig. 3A,B). Conversely, a marked increase in arginine was noted (Fig. 3C). As for the TCA cycle, glutamate, a precursor of α-ketoglutarate, and glutamine were both significantly increased in the naive state (Fig. 3C), while P5C synthase (P5CS) and pyrroline-5-carboxylate dehydrogenase (P5CDH) did not substantially change between two pluripotent states (Fig. 3A). Collectively, through a comparative analysis of naive and primed hESCs, we identified the significantly altered amino acids and metabolic enzymes in proline metabolic axis (Fig. 3D).

To further explore the role of PRODH in the metabolism of hESCs, we employed another untargeted metabolomics to identify alterations in metabolites between naive and primed H9 cells following PRODH KD in each cell type. Based on KEGG and manual screening, we have refined the dataset by excluding all metabolites that are of non-human origins and that are unlikely to be associated with the human embryonic developmental stage. Principal-component analysis (PCA) was performed for the 113 identified metabolites (Datasets EV2,3), which did not demonstrate a distinct segregation between H9-PLKO.1 and H9-shPRODH in the two pluripotent states (Fig. EV3A,B). Out of the 113 detected metabolites, 18.6% (21) showed significant variation (i.e., $P < 0.05$, Student's t-test; fold change cutoff: >1.5 or <0.67) in naive hESCs and 15% (17) in primed hESCs (Fig. EV3C,D). Then, the principal metabolites that significantly changed were shown in the heatmaps (Fig. EV3E,F), mainly organic acids and derivatives. As summarized above, proline and glutamine are two metabolites primarily connected to PRODH. Although there was a trend of increased proline levels and decreased glutamine levels in shPRODH cells, the difference did not reach statistical significance (Fig. 3E). These results suggest that there should be compensatory mechanisms in hESCs to attenuate the effect of PRODH reduction on metabolism.

Further, to assess the effect of proline on naive hESCs, we added an excess of L-proline to Rset and PXGL media. As a result, the proliferation curve of cells cultured in the presence of L-Pro overlapped with that of untreated cells (Fig. 3F). Moreover, the

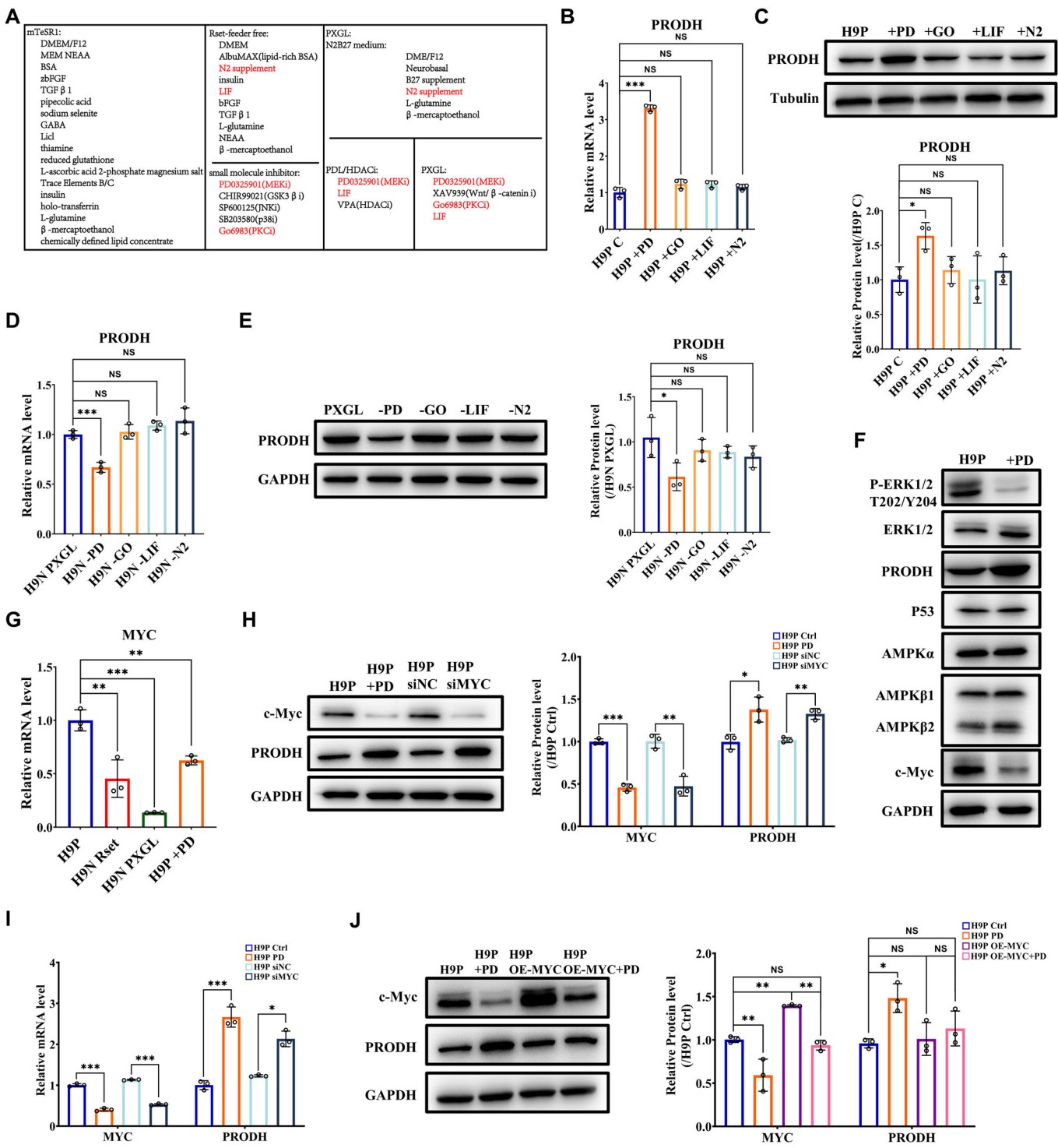

expression levels of NANOG and SOX2 fluctuated in cells supplemented with L-proline at concentrations ranging from 0 mM to 2.5 mM (Fig. 3G), and the trends were unlike those observed after PRODH KD alone (Fig. EV1B,C). These results imply that PRODH did not specifically influence proliferative capacity and pluripotency in the naive state by directly regulating the levels of amino acids, such as proline.

## PRODH knockdown upregulates mtOXPHOS in naive hESCs

Given the little indication that the impairment of naive pluripotency was mediated by the canonical role of PRODH in proline metabolism, our attention was drawn to mtOXPHOS by the KEGG analysis. We found the most enriched pathway in naive cells after

◀ **Figure 2. PRODH is upregulated by PD0325901 in naive hESCs via suppressing MYC.**

(A) Supplemented components in mTeSR1, Rset-feeder free, and PXGL culture systems were listed. Four essential components shared by the Rset and PXGL systems but absent in mTeSR1 were marked in red. (B, C) Additional PD0325901 (PD, 1 μM), Gö 6983 (GO, 2 μM), LIF (10 ng/ml) and N2 (1%) were separately added into mTeSR1, and hESCs were cultured for another 48 h. The mRNA (B) and protein (C) levels of PRODH were determined by qRT-PCR and WB, respectively. Quantifications of the bands relative to α-tubulin and normalized to Control cell values were shown (C). (D, E) The PD0325901, Gö 6983, LIF, and N2 was withdrawn from the PXGL system, respectively. Then, PXGL naive hESCs were cultured for another 48 h. The mRNA (D) and protein (E) levels of PRODH were determined by qRT-PCR and WB, respectively. Quantifications of the bands relative to GAPDH and normalized to H9N PXGL values were shown (E). (F) Primed H9 cells were treated with DMSO (H9P) or PD0325901 (+PD, 1 μM) for 48 h. Cells were harvested and the whole-cell lysates were subjected to SDS-PAGE and WB, and detected by specified antibodies. (G) The mRNA levels of MYC in primed (H9P) cells, Rset (H9N-Rset), and PXGL (H9N-Rset) cultured naive cells, and primed cells treated with 1 μm PD0325901 (H9P + PD) were determined by qRT-PCR. (H, I) After being transfected with siMYC or treated with PD0325901 (1 μm) for 48 h, the expression levels of PRODH in primed H9 cells were analyzed by WB (H) or qRT-PCR (I). Quantifications of the bands relative to GAPDH and normalized to Control cell values were shown (H). (J) Primed H9 cells were infected with lentivirus to overexpress MYC robustly. After 2 days treatment with PD0325901 (1 μM), cells were harvested. The protein levels of MYC and PRODH were determined by WB. Quantifications of the bands relative to GAPDH and normalized to H9P cell values were shown. Data information: The statistical significance was analyzed using unpaired two-tailed Student's t-test in panels (B, C, D, E, G, H, and I) and one-way ANOVA with Bonferroni's multiple comparisons test in panel (J). NS, not significant ($P > 0.05$). *$P < 0.05$, **$P < 0.01$, ***$P < 0.001$. The data shown were all from three independent biological replicates. Each data point represents an independent biological replicate. Data were presented as mean ± SD. Source data are available online for this figure.

PRODH KD being the citric acid cycle, while corresponding pathways in primed cells were pantothenate and CoA biosynthesis (Fig. 4A). Four metabolites (fumaric acid, malic acid, succinic acid, and cis-aconitic acid), constituting 20% of the TCA pathway, were significant increased after PRODH KD in the naive state, while no such effects were observed in primed cells (Fig. 4B). Concomitantly, studies have reported that PRODH is located in the inner mitochondrial membrane and transfers electrons to cytochrome C in the respiratory chain via flavin adenine dinucleotide (FAD) (Phang, 2019). Taken together, we speculated that PRODH might influence naive pluripotency by regulating mtOXPHOS, an energy metabolism that proved to be significantly attenuated in the primed state (Tsogtbaatar et al, 2020; Zhang et al, 2018; Zhou et al, 2012). This may explain why PRODH is specifically associated with naive pluripotency but has little impact on primed pluripotency. Thus, we examined the mitochondrial respiratory capacity of naive and primed hESCs. In the primed state, PLKO.1 or shPRODH cells showed minimal or no increase in OCR (oxygen consumption rate), indicating that their basal and maximal respiratory capacities remained unaltered (Fig. 4C,F). However, in Rset and PXGL naive states, it was noted that the basal and maximal respiratory capacities of shPRODH cells were higher than in control cells, suggesting activated mtOXPHOS (Fig. 4D,E,G,H). Meanwhile, both Rset and PXGL naive cells displayed a decrease in glycolysis or glycolytic capacity with PRODH KD. In contrast, primed cells did not exhibit any changes (Fig. EV4A–F). Consistent with the above results, we also revealed that with shPRODH KD, Rset and PXGL cells exhibited about a 1.5-fold increase in ATP production compared to the control group (Appendix Fig. S3).

As cell differentiation also generated matured mitochondria by promoting mitochondrial fusion and elongation (Seo et al, 2018), we wondered if the enhanced oxidative respiration was due to diminished naive pluripotency. Thus, we performed Mito-Tracker staining to examine the mitochondrial morphology in both primed and naive hESCs following PRODH KD. However, we did not observe significant differences in the mitochondrial morphology between H9-plko.1 and H9-shPRODH cells, as both exhibited granular mitochondrial structures in the naive state (Fig. EV4G). To comprehensively demonstrate the change in mitochondrial length, we used a statistical method, mitochondrial network analysis, which can analyze the branched and complex mitochondrial networks (Valente et al, 2017). Our analysis revealed no significant changes in the mean length of mitochondria following PRODH KD, indicating that the decreased naive cell pluripotency is not the cause of the increased mitochondrial activity (Fig. EV4H). These results imply that a reduction in PRODH activity can specifically enhance the level of mitochondrial oxidative phosphorylation in naive hESCs.

## PRODH maintains naive pluripotency by limiting ROS production

Next, we measured the levels of reactive oxygen species (ROS), which can be elevated through activation of the electron transport chain (ETC) in mitochondria. ROS levels were elevated in H9N-Rset-shPRODH and H9N-PXGL-shPRODH cells compared to PLKO.1 cell, in line with a hyperactive mtOXPHOS (Fig. 5A). Whereas low levels of ROS are essential for cellular function and survival signaling, excessive ROS can cause irreversible oxidative damage to lipids, proteins, and DNA, leading to apoptosis and autophagy (Bigarella et al, 2014; Chang et al, 2022). Thus, we first monitored DNA double-strand breaks using well-established markers, phosphorylated histone H2AX (γH2AX).

The naive hESCs with PRODH KD exhibited an increased level of γH2AX compared with the control cells (Fig. 5B). Additionally, PRODH KD led to a higher incidence of apoptosis in the naive hESCs compared to their primed counterparts (Fig. EV5A). These observations were consistent with the high levels of ROS present in naive hESCs. Further, autophagy in hESCs was analyzed by determining LC3-II/LC3-I ratio and CYTO-ID autophagy detection kit 2.0. Immunoblot analysis showed H9N-shPRODH cells in two culture media had a higher LC3-II/LC3-I ratio than that of H9N-PLKO.1 cells, and the LC3-II enrichment extent in naive hESCs was significantly greater than that in primed hESCs, indicating a higher basal autophagic flux in the naive state (Fig. 5B). Besides, flow cytometric analysis and confocal microscopy revealed higher fluorescence intensity of autophagic vacuoles stained by the CYTO-ID autophagy detection kit 2.0 in H9N-shPRODH cells (Fig. 5C,D). As the primary source of ROS within the cell, mitochondria are highly susceptible to oxidative stress and tend to activate autophagy in response to high levels of ROS (Filomeni et al, 2014). We found an increased co-localization of MitoTracker

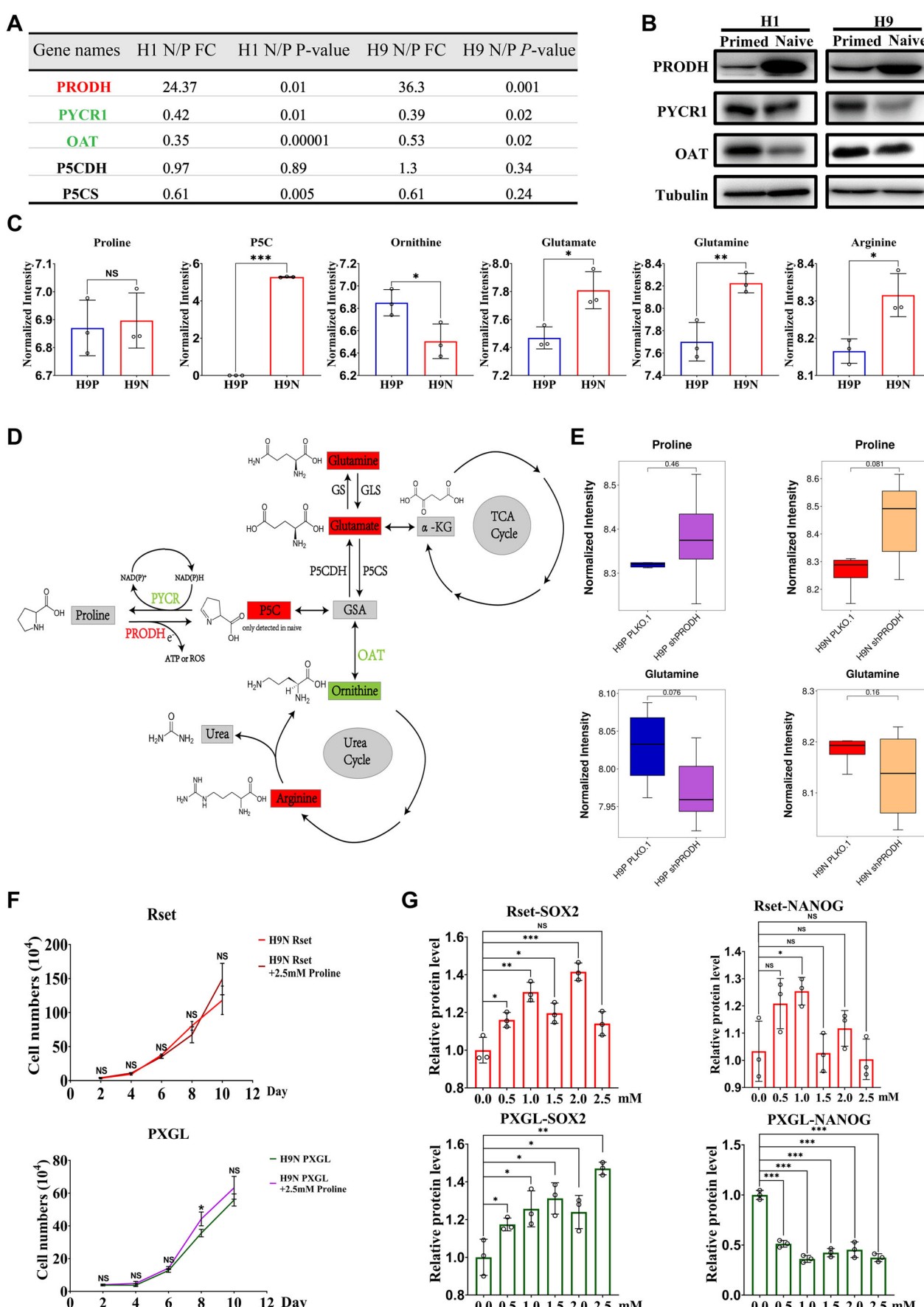

**Figure 3.    Role of PRODH in naive pluripotency regulation is not mediated by proline metabolism.**

(A) Proteomic data of the key kinases associated with proline metabolic pathway were compared between Rset naive and primed H9 hESCs. Kinases marked in red or green were upregulated or downregulated in the naive state, respectively. FC, fold change. (B) The altered protein levels of the key kinases detected in proteomics were confirmed in both H1 and H9 hESCs by WB. (C) Comparison of the key metabolites in proline metabolic pathway between Rset naive (H9N) and primed (H9P) H9 hESCs. Normalized intensity, log10 (scale+1). See also Fig. EV2. (D) Summary of the levels of key components in proline metabolic pathway. Kinases and metabolites marked in red or green were increased or decreased in the naive state, respectively. (E) Comparison of the levels of proline and glutamine between PLKO.1 and shPRODH hESCs in both Rset naive and primed H9 cells. In the box plot, horizontal line represents median, box ranges represent values between quartiles 1 and 3, and whiskers represent values from minimum to maximum. Six biological replicates were analyzed. (F) $8 \times 10^4$ of Rset and PXGL naive H9 hESCs were seeded in 12-well plates and treated with 2.5 mM proline. Cell numbers were counted every 48 h. Cell proliferation curves were shown. (G) Rset and PXGL naive H9 hESCs were treated with varying concentrations of proline for 6 days and were harvested for WB. The band intensities of SOX2 and NANOG were quantitated using ImageJ software and normalized to the values of untreated naive cells. Data information: The statistical significance was analyzed using unpaired two-tailed Student's t-test in panels (C, E, F, and G). NS, not significant ($P > 0.05$). *$P < 0.05$, **$P < 0.01$, ***$P < 0.001$. The data shown were all from atleast three independent biological replicates. In (F and G), each data point represents an independent biological replicate. Data were presented as mean ± SD. Source data are available online for this figure.

Green for mitochondria and LysoTracker Deep Red for lysosomes in Rset-shPRODH and PXGL-shPRODH cells compared to the control group (Appendix Fig. S4A). Pearson's correlation coefficient (PCC) was used to quantitatively analyze the co-localization of two fluorescence by Colocalization Finder, a plugin in ImageJ. After PRODH KD, PCC values greater than 0.5 in Rset and PXGL further confirmed the co-localization of mitochondria and lysosomes, indicating that mitophagy was significantly enhanced (Appendix Fig. S4B). Further, we conducted transmission electron microscopy (TEM) analysis to confirm mitophagy activation at the ultrastructural level. Mitophagosomes–autophagic vacuoles (AVs)-like structures engulfing or containing abnormal mitochondria were observed more frequently in Rset-shPRODH and PXGL-shPRODH cells than in the control group. Quantification of mitophagosomes confirmed that PRODH KD could significantly enhance mitophagy in naive cells (Figs. 5E and EV5B).

In summary, by modulating ROS levels, PRODH significantly impacts critical cellular processes such as DNA damage, apoptosis, and autophagy, which could decrease proliferation and impair pluripotency in hESCs (Cho et al, 2014; Liu et al, 2013; Pan et al, 2013; Vitale et al, 2017; Wilson et al, 2010). Besides, ROS may also directly alter the fate of stem cells through metabolic processes and various signaling pathways (Bigarella et al, 2014). Given these findings, we were curious if the proliferative capacity and pluripotency of H9N-shPRODH cells can be reverted by reducing the production of ROS. After a pre-treatment of 3 h with 50 nM of MitoQ, a mitochondria-targeted antioxidant, Rset-shPRODH and PXGL-shPRODH cells were cultured for an additional 12 h. Upon three hours of MitoQ pre-treatment, naive-shPRODH cells showed a reduction in reactive oxygen species (ROS) levels (Fig. EV5C), alongside an increase in the expression of NANOG and SOX2 (Fig. 5F). The fluorescence of EDU staining demonstrated a rise in Rset-shPRODH and PXGL-shPRODH cells, suggesting a partial restoration in their proliferative capacity (Fig. 5G and Appendix Fig. S5). These results imply that PRODH KD-mediated hyperactive mtOXPHOS leads to ROS accumulation, which ultimately impairs naive pluripotency.

## PRODH facilitates proteolytic degradation of the ETC complex components

Furthermore, we attempted to elucidate the mechanism through which PRODH modulates oxidative phosphorylation activity.

Previous studies have shown that PRODH is linked to Complex II of the mitochondrial electron transport chain in mitochondrial inner membranes (Phang, 2019) and its expression resulted in a dose-dependent down-regulation of Complexes I–IV of the ETC in human colorectal adenocarcinoma epithelial cells (Hancock et al, 2015). To investigate the potential impact of PRODH on the ETC in hESCs, we determined the levels of ETC component proteins, normalized by mitochondrial DNA copy number, in PLKO.1 and shPRODH cells (Fig. 6A–C). The mitochondrial DNA (mtDNA) content was approximately 0.8 times lower than that of the control group upon PRODH KD in Rset and PXGL cells, suggesting a decrease in the number of mitochondria (Fig. 6A). This reduction in mitochondria is consistent with the enhanced mitophagy described above (Figs. 5E and EV5B). Importantly, it was noted that decreasing PRODH/POX expression increased protein levels of the NDUFA10 (subunit of Complex I), SDHA and SDHB (subunits of Complex II), the CIII-R/UQCRFS1 (subunit of cytochrome C reductase, Complex III), and COX IV (subunit of cytochrome C oxidase, Complex IV) not only in naive hESCs but also in their primed counterparts (Fig. 6B,C). However, consistent with the low respiratory activity of primed hESCs and active mtOXPHOS in naive hESCs, all ETC component proteins, especially NDUFA10 and the UQCRFS1, were significantly more expressed in the naive than in the primed state (Fig. 6B). Despite the elevation of expression level, both NDUFA10 and UQCRFS1 in H9P-shPRODH cells remained significantly lower than in normal naive hESCs, which may account for the consistently low levels of oxidative phosphorylation observed in H9P-PLKO.1 and H9P-shPRODH cells. Since Complexes I and III mainly produce ROS among ETC complexes (Tahara et al, 2009), a significant boost in NDUFA10 and UQCRFS1 may contribute to the large increase in ROS in Rset-shPRODH and PXGL-shPRODH cells (Fig. 6B,C). To further explore the molecular connection between ETC and ROS, hESCs were pretreated with MG-132, a specific and cell-permeable proteasome inhibitor, to increase ETC protein level. After incubating with MG-132, Rset-PLKO.1 and PXGL-PLKO.1 hESCs with higher levels of ETC proteins did exhibit higher ROS levels, indicating a strong positive correlation between the levels of ETC and ROS (Fig. 6D). Moreover, PRODH did not appear to affect Complexes I-IV through transcriptional regulation, as the protein levels of these complexes were inconsistent with their corresponding transcripts (Fig. 6E).

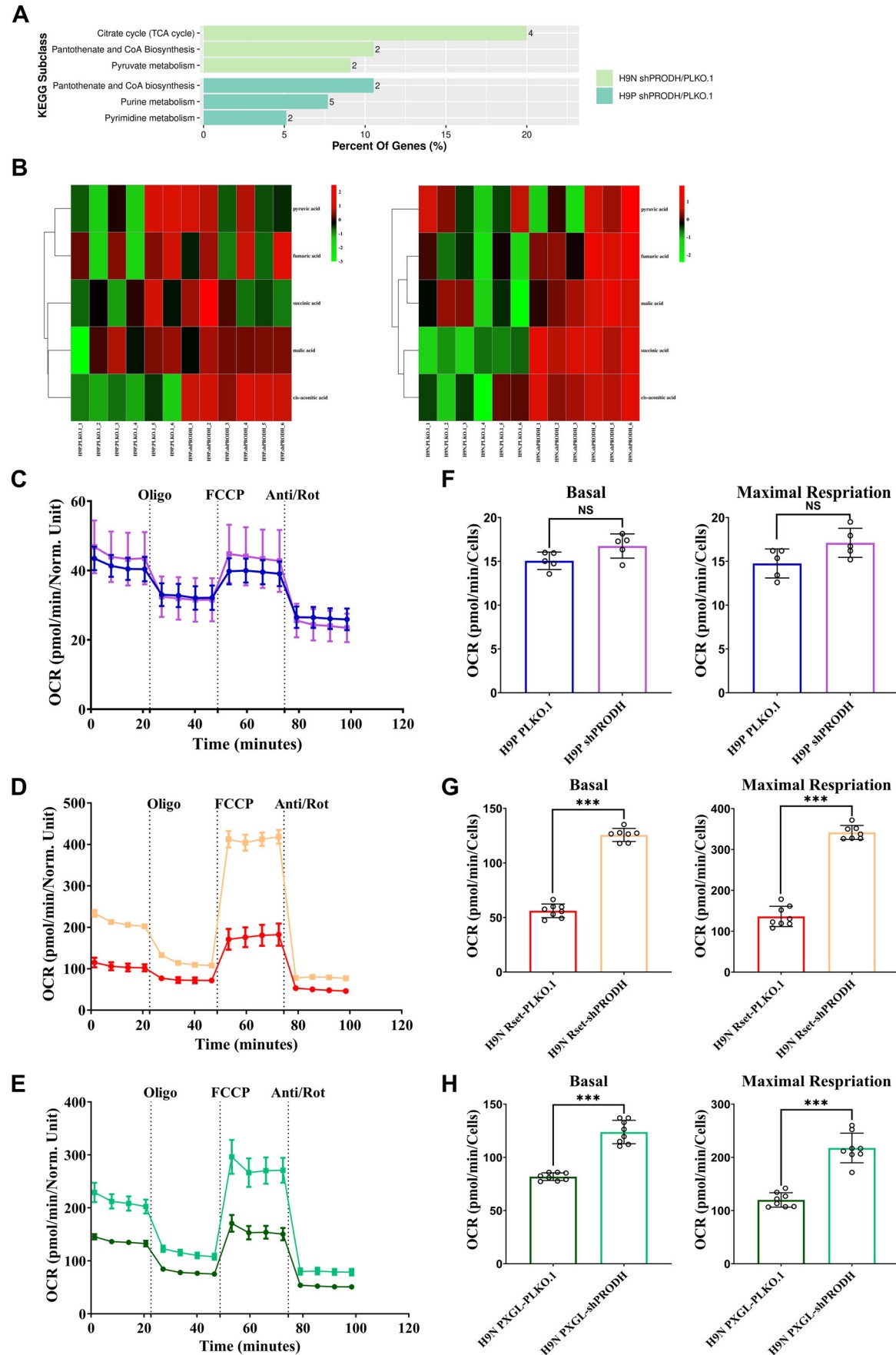

◄ **Figure 4. PRODH knockdown enhances mtOXPHOS in naive hESCs.**

(A) KEGG pathway enrichment analysis for significantly altered metabolites between H9N-shPRODH and H9N-PLKO.1 hESCs (upper), as well as H9P-shPRODH and H9P-PLKO.1 hESCs (lower). The numbers in the figure indicate the count of metabolites enriched in the pathway. (B) Heatmap showing differences in TCA cycle metabolites levels between PLKO.1 and shPRODH H9 hESCs in both primed and Rset naive states. (C–E) Oxygen consumption rates (OCRs) were determined by Seahorse XFe96 extracellular flux analyzer in hESCs cultured in three different media conditions: mTeSR1 (C), Rset (D), or PXGL (E). The cells were treated sequentially with oligomycin (Oligo), carbonyl cyanide 4-(trifluoromethoxy) phenylhydrazone (FCCP), and rotenone (Rot) plus antimycin (AA). Ten technical replicates were analyzed. See also Fig. EV4A–C. (F–H) OCRs were detected and analyzed in hESCs cultured in mTeSR1 (F), Rset (G), and PXGL (H). Basal respiration was calculated as the difference between basal OCR and non-mitochondrial OCR. Maximal OCR was determined as the OCR after FCCP minus non-mitochondrial OCR. Data information: The statistical significance was analyzed using unpaired two-tailed Student's t-test in panels. NS, not significant ($P > 0.05$). ***$P < 0.001$. Data were presented as mean ± SD. In (F–H), each data point represents a technical replicate. Source data are available online for this figure.

We then asked if PRODH regulated the protein stability of the ETC complexes. Naive hESCs were treated with cycloheximide (CHX) to block the synthesis of new proteins, and total endogenous Complexes I-IV in naive PLKO.1 cells were decreased rapidly. In contrast, Complexes I-IV had a much slower degradation rate in shPRODH cells (Fig. 6F,G), indicating that PRODH is required for the rapid degradation of ETC complex proteins. Further, hESCs were pretreated with MG-132 to determine if this degradation occurred via the proteasome-dependent pathway. Prior treatment of the cells with MG-132 for 2 hr before CHX treatment resulted in varying but mild degree of accumulation of the Complexes I-IV proteins, indicating that PRODH-promoted degradation of the ETC complex proteins occurs partially via the proteasome pathway (Fig. 6F,G). It is noteworthy that the tubulin content in MG-132-treated cells shows a notable increase (Wang et al, 2006). Consequently, the actual protein increment of the ETC complex in individual cells treated with MG-132 was considerably higher than that reflected in Western blot whose sample loading was adjusted by tubulin.

Collectively, our results suggest that PRODH KD prevents the proteolysis of the ETC complex proteins, which helps to stabilize those proteins and may lead to more efficient electron transfer to enhance respiration.

## Discussion

Our study herein reveals negative regulation of the ETC components by PRODH in both naive and primed hESCs. The significantly higher mitochondrial activity in naive relative to primed hESCs resulted in preferential regulation of the naive state by PRODH. PRODH highly expressed in naive hESCs limits the activity of the ETC complexes and hence mtOXPHOS, which prevents excessive ROS production, ensuring the maintenance of their stemness and proliferation. We also demonstrate that PD0325901, an essential MEK inhibitor in naive culture medium, critically upregulates PRODH by reducing MYC, revealing a novel mechanism by which MEK/ERK pathway maintains naive pluripotency.

ESCs offer a valuable model for investigating embryonic development and pathogenesis, owing to their ability to proliferate infinitely in vitro and differentiate into nearly all types of adult cells (Loh et al, 2015). Notably, ESCs in the naive state demonstrate a superior ability to generate chimeras compared to primed ESCs (Chen et al, 2015; Wu et al, 2017). naive hESCs can induce differentiation into trophoblast ectoderm and serve as an excellent starting material for studying the pre-implantation blastocyst stage (Guo et al, 2021; Io et al, 2021). Moreover,

emerging evidence suggests that pluripotency is not a discrete, binary state but rather a continuum of cellular states, all possessing the ability to self-renew and differentiate into various cell types while also exhibiting distinct metabolic and mitochondrial characteristics (Tsogtbaatar et al, 2020). Naive state stem cells have bivalent metabolism (glycolysis and OXPHOS), with mitochondria appearing round and displaying sparse and irregular cristae. In contrast, primed stem cells metabolize via glycolysis with mitochondria appearing more elongated shape and containing well-defined transverse cristae (Weinberger et al, 2016; Zhou et al, 2012). It has been reported that the conversed low complex IV cytochrome c oxidase (COX) family transcripts in primed ESCs lead to low mitochondrial respiration (Zhou et al, 2012). Our study showed a significant decrease in the expression levels of all ETC component proteins in the primed state compared to the naive state. However, when considering the transcript levels, we found that while the mRNA levels of SDHB, UQCRFS1, and COX4I1 were notably elevated in the naive cells, the transcripts of NDUFA10 and SDHA did not exhibit a similar increase, particularly in the PXGL naive induction system. Likewise, although PRODH KD resulted in the upregulation of ETC component proteins, their corresponding mRNAs were down-regulated in both naive culture systems. Such inconsistency in the protein versus transcript level indicates a regulatory mechanism of the ETC complex by PRODH. We subsequently showed that PRODH KD could increase the protein stability of the ETC components. A similar phenomenon was reported in colorectal cancer cells, where researchers added DOX to DLD-POX cells to enhance PRODH activity, and the expression levels of all Complex I-IV were reduced. This process could be reversed by antioxidant and PRODH inhibitors (Hancock et al, 2015). Taken together, we speculated that PRODH might directly or indirectly influence oxidative post-translational modifications of ETC components (Lennicke and Cochemé, 2021), thereby regulating their protein stability. Further investigation is warranted to test this hypothesis.

Besides the well-established role in energy metabolism, mitochondria also have a signaling function. Various forms of stress in the mitochondria can generate retrograde signals, such as ROS, that affect other cellular sites and influence stem cell activity and function (Sun et al, 2016). Low levels of ROS provide stem cells with an in vivo quiescent environment necessary for long-term self-renewal, proliferation, and differentiation (Jang and Sharkis, 2007). Excessive ROS, however, can disrupt the regular biological activity of enzymes and transcription factors (Ji et al, 2010; Lennicke and Cochemé, 2021) and damage various organelles by impairing proteins, lipids, or even DNA, ultimately inducing autophagy and apoptosis (Nugud et al, 2018). Consistent with these data, our

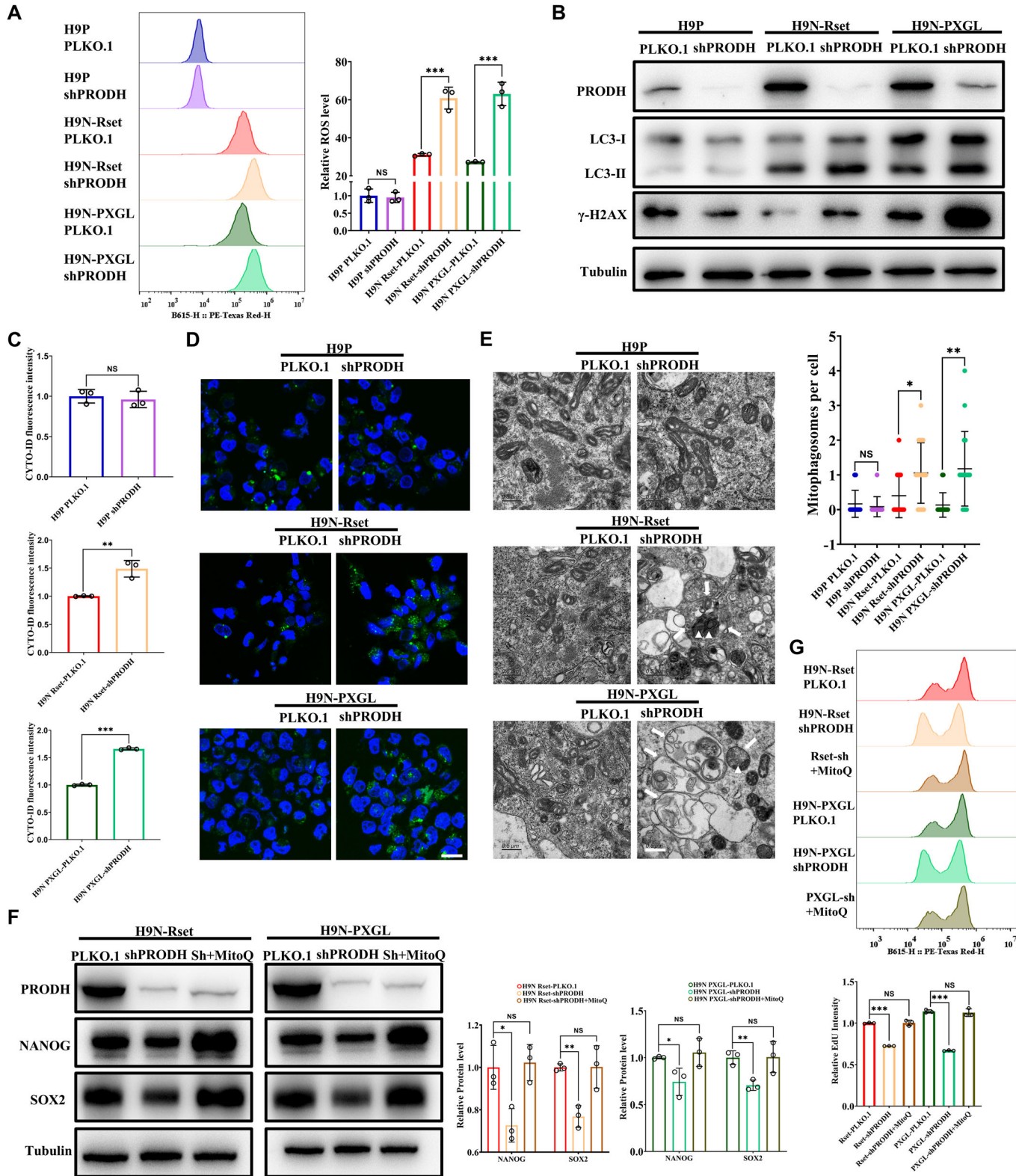

**Figure 5. PRODH maintains naive pluripotency by limiting ROS production.**

(A) ROS levels were determined by MitoSOX. Fluorescent intensity of cells stained with MitoSOX was determined by flow cytometry (left), and quantitative data normalized to H9P PLKO.1 values were presented (right). (B) Primed and naive (Rset and PXGL) hESCs were harvested, and the whole-cell lysates were subjected to SDS-PAGE and WB, and detected by specified antibodies. (C, D) Autophagic vacuoles (AV) were detected by CYTO-ID autophagy detection kit 2.0. Flow cytometry determined the fluorescent intensity of cells stained with CYTO-ID, and quantitative data normalized to H9P PLKO.1 values were presented (C). Representative confocal micrographs were shown in Fig. 4D. Scale bars, 20 µm. (E) Primed and naive (Rset and PXGL) hESCs were harvested for examination by transmission electron microscopy (TEM). Representative TEM images (left) and quantitative analysis (right) showed a striking increase in autophagic vacuoles (AVs) and mitophagosome-like structures – AVs containing engulfed mitochondria after PRODH KD in naive states. The white arrows indicate AV-like organelles and the white triangles indicate mitochondria enclosed within AVs. Scale bars, 0.5 µm. More TEM images captured at various magnification levels were presented in Fig. EV5B. (F) Rset-shPRODH and PXGL-shPRODH cells were treated with 50 nM MitoQ for 3 h, and cultured for an additional 12 h. Cells were harvested and the whole-cell lysates were subjected to SDS-PAGE and WB, and detected by specified antibodies. Quantifications of the bands relative to α-tubulin and normalized to Control cell values (H9N Rset-PLKO.1 or H9N PXGL-PLKO.1) were also shown (right). (G) Rset-shPRODH and PXGL-shPRODH cells were treated with 50 nM MitoQ for 3 h, and cultured for an additional 12 h. The cell proliferation rate represented by EdU incorporation was assessed using EdU assay. The fluorescent intensity of cells stained with EdU was determined by flow cytometry (upper), and the quantitative data normalized to Control cell values (H9N Rset-PLKO.1 or H9N PXGL-PLKO.1) was shown (lower). Representative images of EdU stain were presented in Appendix Fig. S5. Data information: The statistical significance was analyzed using unpaired two-tailed Student's t-test in panels (A, C, E, F, and G). NS, not significant ($P > 0.05$). *$P < 0.05$, **$P < 0.01$, ***$P < 0.001$. In (A, C, F, and G), the data shown were from three independent biological replicates, and each data point represents an independent biological replicate. In (E), a total of 12 cell images for H9P PLKO.1, 12 cell images for H9P shPRODH, 15 cell images for H9N Rset-PLKO.1, 18 cell images for H9N Rset-shPRODH, 15 cell images for H9N PXGL-PLKO.1 and 17 cell images for H9N PXGL-shPRODH were analyzed. Each data point represents the number of mitophagosomes derived from each cell image. Data were presented as mean ± SD. Source data are available online for this figure.

findings revealed that activated mtOXPHOS increased ROS levels in naive shPRODH cells, resulting in DNA damage, autophagy, and apoptosis. Significantly, MitoQ restored pluripotency and cell proliferation in the naive state by reducing ROS levels, highlighting the importance of maintaining appropriate ROS levels in naive hESCs, and the crucial protective role of PRODH in preventing excessive ROS production.

L-proline metabolism is emerging as a key pathway in regulating pluripotency and differentiation (Kilberg et al, 2016; Washington et al, 2010). Supplementation of mESC culture media with proline is sufficient to promote cell proliferation and transition of mESCs to EpiSCs (Casalino et al, 2011; Washington et al, 2010) and induce ESCs into a mesenchymal-like, motile phenotype, which can be reversed using vitamin C that promotes the demethylation of H3K9 and H3K36 (Comes et al, 2013). Our study provided a comprehensive landscape of the differences in the proline metabolic axis between naive and primed hESCs. The active TCA cycle is supported by glutamine oxidation and alpha-ketoglutarate (α-KG) converted from glutamate (Yang et al, 2014; Yoo et al, 2020). High glutamate and glutamine levels may benefit unique active mtOXPHOS in the naive state. Besides, glutamine also participates in the biosynthesis of nucleotides, glutathione (GSH), and other nonessential amino acids (Yoo et al, 2020). Some reports have suggested that glutamine metabolism is essential to regulate the degradation of OCT4, and glutamine-dependent signaling may help coordinate the differentiation of the three germ layers (Lu et al, 2022; Marsboom et al, 2016). In future study, modeling of changes in extracellular metabolites and employing isotopic tracers in conjunction with Mass Spectrometry (MS) or Nuclear Magnetic Resonance (NMR) would be important to map out the metabolite fluxes of proline metabolic pathway. Unexpectedly, in this study, it was found that PRODH, the mitochondrial enzyme responsible for catalyzing the initial step in proline catabolism, did not influence naive pluripotency by modulating the cellular levels of proline or glutamine. No significant changes were observed in proline and glutamine levels after PRODH KD, indicating the presence of a resilient compensatory mechanism that effectively maintains metabolic homeostasis in the naive state. Direct measuring,

comparing and manipulating the enzymatic activity of PRODH in naive hESCs versus primed hESCs may help to further clarify if and to what extent the canonical role of PRODH contributes to maintaining human naive pluripotency.

Further, we showed that the MEK inhibitor PD0325901 decreased MYC in naive hESCs that, in turn, promoted PRODH transcription. PD0325901 is widely used in establishing and maintaining human naive pluripotent stem cells (Bredenkamp et al, 2019; Gafni et al, 2013; Takashima et al, 2014; Theunissen et al, 2014). Conversely, activation of MEK/ERK signaling promotes the differentiation of ESCs. However, the mechanism by which MEK/ERK signaling pathway regulates naive pluripotency is only partly understood. Some reports have demonstrated the inhibitory effect of MEK/ERK on MYC (Ciccarelli et al, 2018; Marampon et al, 2006). MYC suppressed POX/PRODH expression primarily through upregulating miR-23b*, which selectively reduces translation by binding to the 3'-untranslated region of the messenger RNA PRODH/POX with a particular sequence (Liu et al, 2012). Interestingly, we also found that the c-Myc expression levels in the shPRODH group were consistently elevated compared to the control group in both naive and primed states (Appendix Fig. S2B). This result further implicated a strong connection between MYC and PRODH that warrants further investigation. Thus, we revealed the mechanism underlying upregulation of PRODH in naive hESCs by the supplemented MEK inhibitor and discovered a novel connection between mitochondria and the MEK/ERK pathway.

Interestingly, PRODH in a human osteosarcoma cell line (U2OS) induced cell senescence associated with the increase in ROS production and accumulation of DNA damage (Nagano et al, 2017), a finding that is in stark contrast to our current finding with naive hESCs. The mechanisms underlying such discrepancy remain unclear but could be related to differential posttranslational modifications of PRODH proteins and their binding partners in different cellular contexts. Clearly, the differential roles and regulatory mechanisms of PRODH in maintaining naive pluripotency versus promoting cell senescence warrant further in-depth investigations.

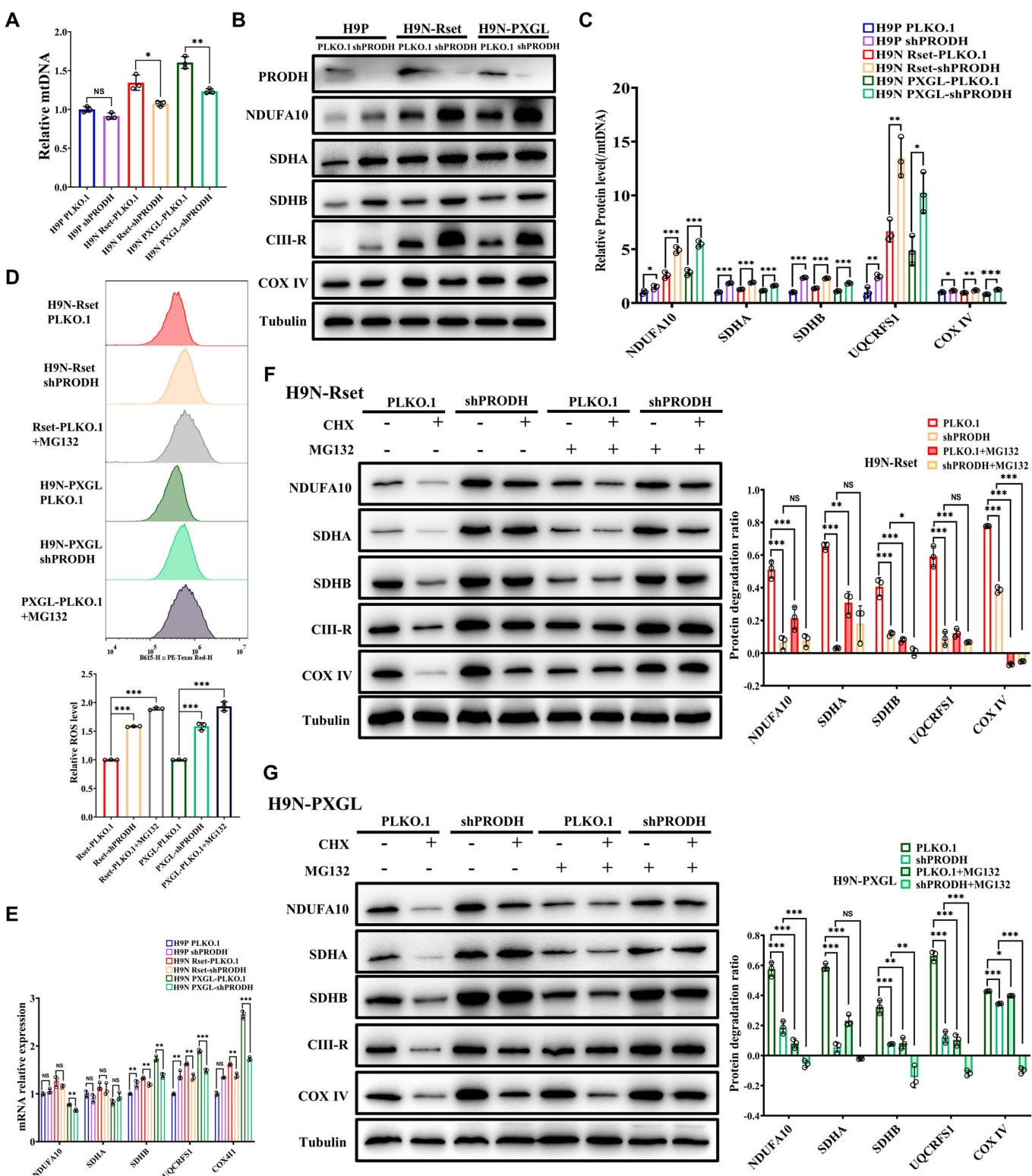

In summary, we revealed a much higher expression level of PRODH in naive hESCs than in their primed counterparts. The high levels of PRODH, induced by the MEK inhibitor PD0325901 present in naive culture media, can facilitate ETC complex protein degradation to limit mtOXPHOS and ROS level in hESCs, and hence safeguarding human naive pluripotency. These results implicate a crucial role of PRODH in regulating mtOXPHOS-dependent ROS production and pluripotency in various stem cells, such as naive pluripotent stem cells, hematopoietic stem cells, neural stem cells, muscle stem cells, and even cancer stem cells.

**Figure 6.  PRODH facilitates the proteolytic degradation of the ETC complex proteins.**

(A) Levels of mitochondrial DNA were measured by qRT-PCR using the Human Mitochondrial DNA Monitoring Primer Set. (B) Primed and naive (Rset and PXGL) H9 hESCs were harvested and the whole-cell lysates were subjected to SDS-PAGE and WB, and detected by specified antibodies. (C) The ETC protein bands in (B) were quantitated using ImageJ software. The mean protein levels of Complex I-IV were normalized to the relative mtDNA copy number (A), with the values of H9P-PLKO.1 being set at 1. (D) Rset-PLKO.1 and PXGL-PLKO.1 hESCs were incubated with 5 μM MG-132 for 24 h. ROS levels of cells stained with MitoSOX were determined by flow cytometry (upper), and quantitative data normalized to Control cell values (H9N Rset-PLKO.1 or H9N PXGL-PLKO.1) were presented (lower). (E) The mRNA levels of ETC component proteins in primed and naive H9 hESCs were determined by qRT-PCR. (F, G) Rset and PXGL naive H9 hESCs were pretreated with 5 μM MG-132 or vehicle for 2 h, followed by addition of 20 μg/ml cycloheximide (CHX) and further incubation of 24 h still in the presence of MG-132. Whole-cell lysates were immunoblotted with the indicated antibodies. Mean protein levels of Complex I-IV were quantitated using ImageJ and normalized by α-tubulin. Protein degradation ratio = (protein level in cells without CHX treatment - protein level in cells treated with CHX)/protein level in cells without CHX treatment. Data information: The statistical significance was analyzed using unpaired two-tailed Student's t-test in panels (A, C, D, and G) and one-way ANOVA with Bonferroni's multiple comparisons test in panels (E and F). NS, not significant ($P > 0.05$). *$P < 0.05$, **$P < 0.01$, ***$P < 0.001$. The data shown were all from three independent biological replicates. Each data point represents an independent biological replicate. Data were presented as mean ± SD. Source data are available online for this figure.

# Methods

## qPCR primer sequences

| Gene name | Forward primer | Reverse primer |
|---|---|---|
| ACTB | AACCGCGAGAAGATGACCCA | GGATAGCACAGCCTGGATAGCA |
| PRODH | CCGCAGGAATGGTGTCATCA | GACTCTACCTGAGGCTTCGAT |
| PRDM14 | GCATACTCCGCACACACATCA | CAGCCATCATCCTCCTTGTGT |
| KLF4 | TCAACCTGGCGGACATCAAC | CAGCACGAACTTGCCCATCA |
| STELLA | CAGCAGGAGAGGAGTAAGAACA | TGAAGTGGCTTGGTGTCTTGA |
| B3GAT1 | CTCCTTCGAGAACTTGTCACC | GGGTCAGTGAAGCCCTTCTT |
| HERVH | GCCTCTGCTCCTCCACCCTATAA | CGTTTAGCTCCAGCCACCTTTTT |
| ZIC2 | GCGCAACTCCACAACCAGTA | TGCCGCATATAGCGGAAAAAG |
| SOX2 | CATGGGTTCGGTGGTCAAGTC | TCGGCGCCGGGGAGATACA |
| NANOG | CTGTGATTTGTGGGCCTGAAGAA | TTTGGGACTGGTGGAAGAA |
| POU5F1 | GTGGAGGAAGCTGACAACAA | ATTCTCCAGGTTGCCTCTCA |
| FOXA2 | GGAGCAGCTACTATGCAGAGC | CGTGTTCATGCCGTTCATCC |
| SOX17 | GTGGACCGCACGGAATTTG | GGAGATTCACACCGGAGTCA |
| GATA4 | TCCCTCTTCCCTCCTCAAAT | TCAGCGTGTAAAGGCATCTG |
| GATA6 | TCCACTCGTGTCTGCTTTTG | TCCTAGTCCTGGCTTCTGGA |
| CK18 | AGCTCAACGGGATCCTGCTGCACCTTG | CACTATCCGGCGGGTGGTGGTCTTTTG |
| AFP | CTTTGGGCTGCTCGCTATGA | GCATGTTGATTTAACAAGCTGCT |
| NKX2.5 | AAGTGTGCGTCTGCCTTTCCCG | TTGTCCGCCTCTGTCTTCTCCA |
| NKX3.1 | CCATACCTGTACTGCGTGGG | TGCACTGGGGGAATGACTTA |
| MSX1 | ACACAAGACGAACCGTAAGCC | CACATGGGCCGTGTAGAGTC |
| MEF2C | TTTAACACCGCCAGCGCTCTTCACCTTG | TCGTGGCGCGTGTGTTGTGGTATCTCG |
| BRACHYURY | TATGAGCCTCGAATCCACATAGT | CCTCGTTCTGATAAGCAGTCAC |
| BMP4 | CACTGGCTGACCACCTCAAC | GGCACCCACATCCCTCTACT |
| MAP2 | CAGGTGGCGGACGTGTGAAAATTGAGAGTG | CACGCTGGATCTGCCTGGGGACTGTG |
| PAX6 | TGGGCAGGTATTACGAGACTG | ACTCCCGCTTATACTGGGCTA |
| HES4 | GAGCGCGTATTAACGAGAGC | GCAGGTGTCTCACGGTCATC |
| NDUFA10 | GTGCAAACTGCGCTATGGAAT | CAGGAAAGTGCTTGAAGCCTA |
| SDHA | CAAACAGGAACCCGAGGTTTT | CAGCTTGGTAACACATGCTGTAT |
| SDHB | ACAGCTCCCCGTATCAAGAAA | GCATGATCTTCGGAAGGTCAA |
| UQCRFS1 | CTGAATACCGCCGCCTTGAA | ATGCGACACCCACAGTAGTTA |
| COX4I1 | CAGGGTATTTAGCCTAGTTGGC | GCCGATCCATATAAGCTGGGA |
| MYC | GCCTCAGAGTGCATCGAC | TCCACAGAAACAACATCG |

## Immunofluorescence antibodies

| Anti-PRODH | Santa Cruz | Cat#SC-376401;RRID:AB_11149907 |
|---|---|---|
| Anti-OCT4A | Cell signaling | Cat#2890; RRID:AB_2167725 |
| Anti-SOX2 | Cell signaling | Cat#2748 s; RRID:AB_823640 |
| Anti-NANOG | Cell signaling | Cat#4903 s; RRID:AB_10559205 |
| Anti-α-Tubulin | GNI Group | Cat#GNI4310-AT |
| Anti-β-Actin | GNI Group | Cat#GNI4310-BA |
| Anti-GAPDH | GNI Group | Cat#GNI4310-GH |
| Anti-PYCR1 | Abcam | Cat#ab279385 |
| Anti-OAT | Abcam | Cat#ab137679;RRID:AB_2924872 |
| Anti-LC3 | Cell signaling | Cat#12741;RRID:AB_2617131 |
| Anti-γ-H2AX | Abclonal | Cat#AP0099;RRID:AB_2771168 |
| Anti-NDUFA10 | Abcam | Cat#ab174829 |
| Anti-SDHA | Abcam | Cat#ab137040;RRID:AB_2884996 |
| Anti-SDHB | Abcam | Cat#ab175225;RRID:AB_2904585 |
| Anti-UQCRFS1 | Abcam | Cat#ab191078;RRID:AB_2687933 |
| Anti-COX IV | Abcam | Cat#ab202554;RRID:AB_2861351 |
| Anti-Phospho-ERK1/2(Thr202/Tyr204) | Cell signaling | Cat#4370 S;RRID:AB_2315112 |
| Anti-ERK1/2 | Cell signaling | Cat#4695 T;RRID:AB_390779 |
| Anti-p53 | Beyotime | Cat#AF5258 |
| Anti-AMPKα | Cell signaling | Cat#5831;RRID:AB_10622186 |
| Anti-AMPKβ | Cell signaling | Cat#4150;RRID:AB_10828832 |
| Anti-C-Myc | TransGen | Cat#HT101;RRID:AB_2832251 |

## Human cell lines and drug treatment

Male and female human (h) ESC lines H1 and H9 (SCSP-301 and 302, respectively) were purchased from the Cell Bank of the Chinese Academic of Sciences in Shanghai, China. Where appropriate, hESCs were treated with Cycloheximide (CHX) (Sigma-Aldrich, #239763M), MG132 (Sigma-Aldrich, M7449), PD0325901 (ABCR, #AB253775), L-proline (Aladdin, #P108709),

MitoQ (MedChemExpress, HY-100116A) either individually or in combination with the indicated concentrations.

## Primed and RSet Feeder-Free (Rset) naive hESC culture

To avoid any potential confounding effects by irradiated mouse embryonic fibroblast (MEF) feeder layers, hESCs used in this study were grown in feeder-free conditions. According to the manufacturer's instructions, primed hESCs were cultivated on Matrigel-coated tissue culture dishes (Corning, #354277) and kept alive in mTeSR1 medium (StemCell Technologies, #85850) at 37 °C and 5% $O_2$. Medium change was performed on a daily basis. Cells were passaged every three to four days using 0.02% EDTA incubated for 5 min at 37 °C (Cienry, #E6511).

In RSeT Feeder-Free (Rset) medium, primed hESCs were transformed into naive-like hESCs by following the instructions provided by the manufacturer (StemCell Technologies, #05975). Thereafter, the culture medium was changed every two days, and naive-like hESCs were passaged using TrypLE (Thermo Fisher, #12605010).

## PXGL naive hESC culture

Conversion of primed hESCs to naive hESCs in the PXGL medium was achieved following the protocol previously published (Bredenkamp et al, 2019; Chen et al, 2022). After initial passages on MEF, stabilized cells can then be maintained on Geltrex (growth factor-reduced, Thermo Fisher, #A1413302) and dissociated with TrypLE. After plating the cell suspension, add 1 μl/cm$^2$ of Geltrex immediately to each well and gently shake to mix it all together. Y-27632, a ROCK inhibitor, was introduced during replating. Cells were fed with new PXGL media every two days. Cells can be split every 5 days at a ratio of 1:3 and cryopreserved using CryoStor CS10 (StemCell Technologies, #07930).

## Reverse transcription quantitative real-time PCR

Total RNA was extracted using TRIzol reagent (Life Technologies, #15596-026), and PrimeScript RT reagent kit with gDNA eraser (TaKaRa, #DRR047S) was used to make complementary DNA (cDNA) from 1 mg of total RNA. We used iTaq Universal SYBR Green Supermix (Bio-Rad, #1725124) and Step One Plus (Applied Biosystems) for real-time PCR. All the PCR amplification was performed in triplicate and repeated in three independent experiments.

## Western blotting (WB)

Collected cell samples were lysed with the Laemmli Sample Buffer (BIO-RAD, #1610747). Ten μL of loading samples were run through a 12% SDS-PAGE before being transferred to 0.45 mm or 0.22 mm polyvinyl difluoride membranes (Millipore, #IPVH00010 or #ISEQ00010). Membranes were probed with the listed primary and corresponding secondary antibodies after being blocked in 5% skim milk. Finally, the blots were developed using the ECL reagent (Share-Bio, #sb-wb012) and visualized by the Tanon 5200 Multi Chemiluminescent Imager system. Quantitation of protein band intensity in immunoblots was conducted by ImageJ software.

## Lentiviral vector construction, viral production, and viral infection

Primed or naive hESCs with stably knock-down PRODH were generated by lentivirus-mediated transduction. A shRNA fragment targeting PRODH was generated using a pair of primers (forward primer: 5′-CCGGGAGGTGCTTCTTTCACCAAATCTCGAG ATTTGGTGAAAGAAGCACCTCTTTTTG-3′ where the target sequence was underlined; reverse primer: 5′-AATTCAAAAAGAG GTGCTTCTTTCACCAAATCTCGAGATTTGGTGAAAGAAGCA CCTC-3′) and cloned into the AgeI and EcoRI sites of the plasmid pLKO.1 puro lentiviral vector (Addgene, #8453) to form the new plasmid "shPRODH". Besides, the scramble sequence 5′-CCGG CCTAAGGTTAAGTCGCCCTCGCTCTAGCGAGGGCGACTTAA CCTTTTTTT-3′ was cloned into the plasmid pLKO.1 as described above and the modified plasmid was designated as "PLKO.1". To overexpress c-Myc, the lentiviral expression plasmid pLVX-MYC was constructed from the cDNA sequence of pCDNA3.1-MYC(human)-3×FLAG plasmid (Miaoling Biology, # P17491) expression vector.

Then, lentiviruses were generated in HEK293T cells (SCSP-502, from the Cell Bank of the Chinese Academic of Sciences) by co-transfecting the HEK293 cells with psPAX2 (Addgene, #12260), pMD2.G (Addgene, #12259), and constructed lentiviral plasmids using Lipofectamine 3000 (Thermo, #L3000015). The medium was changed to Opti-MEM containing 5% FBS and 1% sodium pyruvate 6 to 8 h after transfection. On day 1 following transfection, the culture medium containing the released recombinant viruses was harvested, fresh media were added, and the culture media were harvested once more on day 2 following transfection. The recombinant virus-containing supernatants were filtered with a 0.45-mm filter, and viral particles were concentrated by centrifugation at 25,000 rpm in a Beckman SW 28 rotor for 100 min at 4 °C. Following a 12-h transduction with concentrated viruses, hESCs were grown in the fresh medium for a further 48 h before blasticidin was introduced.

### Plasmid and siRNA transfection

Primed hESCs were digested into single-cell suspension and seeded in 12-well plates with ROCK inhibitor Y27632 treatment at a density of $1–1.5 × 10^5$ cells/well. After 24 h, cells were starved for 3 h in Opti-MEM (Thermo, #31985062), and then, following the manufacturer's instructions, Lipofectamine RNAiMAX Reagent (Thermo, #13778075) was used to transfect the cells with siRNAs (siMYC: 5′-GAGGAGACATGGTGAACCA-3′; scramble siRNA served as the control: 5′-GGCTCTAGAAAAGCCTATGC-3′). After six hours, the medium was rejuvenated. Cells were cultured for an additional 48 h before being collected for determining mRNA or protein levels.

## Alkaline phosphatase staining

To assess the pluripotency of primed and naive hESCs after knocking down PRODH, all PLKO.1 and shPRODH cells were digested into single cells and plated at a density of 50,000 cells per well in 6-well plates without ROCK inhibitor Y27632 treatment and cultured for 5 days. The hESC colonies were subsequently stained using the alkaline phosphatase (AP) staining kit (C3250S, Beyotime Biotechnology). Representative cellular morphology images were

captured, and the number of positive colonies was calculated using ImageJ (V.1.54d).

## Immunofluorescence microscopy

After being washed twice with phosphate-buffered saline (PBS), the hESCs were fixed with 4% formaldehyde (Sigma, #F8775) in PBS for 15 min at room temperature and permeabilized with 0.2% Triton X-100 (Diamond, # A110694) in PBS at room temperature for 10 min. They were then blocked with 3% Bovine serum albumin (BSA, Sigma, #A9418) in PBS for 45 min after being rinsed three times with PBS. Cells were treated with the appropriate primary antibodies for overnight incubation at 4 °C and the secondary antibodies for an hour at room temperature, with three washes in between. Finally, the cell samples were visualized using a Nikon A1R confocal microscope. The fluorescence quantification was performed by ImageJ (V.1.54d).

## EdU (5-Ethynyl-2′-deoxyuridine) incorporation assay

hESCs were washed with PBS. Fresh medium with 10 µM EdU was then added, and incubation continued for 2 h. After the incubation, cells were fixed with 4% formaldehyde and washed with PBS containing 3% BSA. Then, hESCs were permeabilized with 0.3% Triton X-100 and stained using the BeyoClick™ EdU Cell Proliferation Kit with Alexa Fluor 594 (C0078S, Beyotime Biotechnology) according to the manufacturer's instructions. Finally, EdU-labeled cells were measured by an ACEA NovoCyteTM flow cytometer, and representative images were examined with fluorescence microscopy.

## Cell proliferation assay

Primed and naive hESCs were seeded in 12-well plates, and proline was added as indicated. Primed hESCs were transformed into naive-like hESCs by following the instructions described above. Cells were digested and counted using a Countess II FL Automated Cell Counter (Thermo). Growth curves were plotted using the GraphPad Prism 9.0 software.

## Untargeted metabolomics

### LC-MS sample preparation

The collected samples were thawed on ice, and metabolite was extracted with 50% methanol Buffer. Briefly, 20 µL of sample was extracted with 120 µL of precooled 50% methanol, vortexed for 1 min, and incubated at room temperature for 10 min; the extraction mixture was then stored overnight at −20 °C. After centrifugation at 4000 × g for 20 min, the supernatants were transferred into new 96-well plates. The samples were stored at −80 °C before the LC-MS analysis. In addition, pooled QC samples were also prepared by combining 10 µL of each extraction mixture (Yu et al, 2018).

### LC-MS/MS analysis

Firstly, all chromatographic separations were performed using a Thermo Scientific UltiMate 3000 HPLC. An ACQUITY UPLC BEH C18 column (100 mm*2.1 mm, 1.8 µm, Waters, UK) was used for the reversed phase separation. The column oven was maintained at

35 °C. The flow rate was 0.4 ml/min and the mobile phase consisted of solvent A (water, 0.1% formic acid) and solvent B (Acetonitrile, 0.1% formic acid). Gradient elution conditions were set as follows: 0–0.5 min, 5% B; 0.5–7 min, 5% to 100% B; 7–8 min, 100% B; 8–8.1 min, 100% to 5% B; 8.1–10 min, 5%B. The injection volume for each sample was 4 µl.

A high-resolution tandem mass spectrometer Q-Exactive (Thermo Scientific) was used to detect metabolites eluted from the column. The Q-Exactive was operated in both positive and negative ion modes. Precursor spectra (70–1050 $m/z$) were collected at 70,000 resolution to hit an AGC target of 3e6. The maximum inject time was set to 100 ms. A top 3 configuration to acquire data was set in DDA mode. Fragment spectra were collected at 17,500 resolution to hit an AGC target of 1e5 with a maximum inject time of 80 ms. In order to evaluate the stability of the LC-MS during the whole acquisition, a quality control sample (pool of all samples) was acquired after every 10 samples.

### Data processing

The acquired MS data pretreatments, including peak picking, peak grouping, retention time correction, second peak grouping, and annotation of isotopes and adducts were performed using the XCMS software. LC–MS raw data files were converted into mzXML format and then processed by the XCMS, CAMERA, and metaX (Li et al, 2018) toolbox implemented with the R software. Each ion was identified by combining retention time (RT) and $m/z$ data. Intensities of each peak were recorded and a three-dimensional matrix containing arbitrarily assigned peak indices (retention time-$m/z$ pairs), sample names (observations) and ion intensity information (variables) was generated. The online Kyoto Encyclopedia of genes and genomes (KEGG), human metabolism database (HMDB) database was used to annotate the metabolites by matching the exact molecular mass data ($m/z$) of samples with those from the database. If a mass difference between the observed and the database value were less than 10 ppm, the metabolite would be annotated, and the molecular formula of metabolites would be further identified and validated by the isotopic distribution measurements. We also used an in-house fragment spectrum library of metabolites to validate the metabolite identification.

The intensity of peak data was further pre-processed by metaX (Li et al, 2018). Those features detected in less than 50% of QC samples or 80% of biological samples were removed, and the remaining peaks with missing values were imputed with the k-nearest neighbor algorithm to improve the data quality further. PCA was performed for outlier detection and batch effects evaluation using the pre-processed dataset. Quality control-based robust LOESS signal correction was fitted to the QC data with respect to the order of injection to minimize signal intensity drift over time. In addition, the relative standard deviations of the metabolic features were calculated across all QC samples, and those >30% were then removed.

Based on KEGG and manual selection, the dataset was further refined by excluding all metabolites that are of non-human origins and that are unlikely to be associated with human embryonic developmental stage. Student' t-tests were conducted to detect differences in metabolite concentrations between two phenotypes. The $P$-value was adjusted for multiple tests using an FDR (Benjamini–Hochberg). The VIP value was calculated. A VIP cut-off value of 1.0 was used to select important features.

### Bioinformatics analysis

Bioinformatic analysis was performed using the OmicStudio tools at https://www.omicstudio.cn/tool (Lyu et al, 2023). Briefly, principal component analysis (PCA) was performed using R to assess the data quality. Only metabolites with fold change >1.5 or <0.67 fold and a *P*-value < 0.05 were further analyzed. Volcano plot analysis was also done in R using the ggplot2 program. Besides, R was also applied to perform hierarchical clustering analysis with the pheatmap package.

The untargeted metabolomics data were deposited to the EMBL-EBI's MetaboLights repository (Haug et al, 2020) under accession code MTBLS7840.

## Targeted metabolomics

### Sample preparation

The samples were thawed at 4 °C and mixed with 1 mL of cold methanol/acetonitrile/$H_2O$ (2:2:1,v/v/v). The homogenate was sonicated at low temperature (30 min/once, twice), and the mixture was centrifuged for 20 min (14,000 × *g*, 4 °C). The supernatant was dried in a vacuum centrifuge. For LC-MS analysis, the samples were re-dissolved in 100 μL acetonitrile/water (1:1, v/v), adequately vortexed, and then centrifuged (14,000 × *g*, 4 °C, 15 min). The supernatants were collected for LC-MS/MS analysis.

### HPLC-MS/MS analysis

Analyses were performed using a UHPLC (1290 Infinity LC, Agilent Technologies) coupled with a QTRAP (AB Sciex 5500). the mobile phase contained A = 25 mM $CH_3COONH_4$ + 0.08% FA in water and B = 0.1% FA in ACN. The samples were in the automatic sampler at 4 °C, and the column temperatures were kept constant at 40 °C. The gradients were at a flow rate of 250 μL/min, and a 2 μL aliquot of each sample was injected. The gradient was 90% B linearly reduced to 70% in 0–12 min, then reduced to 50% in 12–18 min, and reduced to 40% in 18–25 min, and kept for 25–30 min. Then the B was increased to 90% in 30–30.1 min and kept for 30.1–37 min. The QC samples used for testing and evaluating the stability and repeatability of this system, at the same time, set the standard mixture of AA metabolites, used for correction of chromatographic retention time.

In ESI positive modes, the conditions were set as follows: source temperature 500 °C, ion Source Gas1 (Gas1): 40, Ion Source Gas2 (Gas2): 40, Curtain gas (CUR): 30, ionSapary Voltage Floating (ISVF) 5500 V; Adopt the MRM mode detection ion pair.

### Data processing

The Multiquant software was used to extract chromatographic peak area and retention time. Use the AA standards' correct retention time to identify the metabolites (Cai et al, 2015). The quality control samples were processed together with the biological samples. Detected metabolites in pooled samples with coefficient of variation (CV) less than 30% were denoted as reproducible measurements.

In order to identify the differential metabolites, statistical analyses between two sample groups were performed by calculating the fold changes and *P*-values of metabolites. Student's t-test was used to obtain *P*-values. Metabolites with *p* values < 0.05 and fold changes >1.5 were marked as the significantly changed metabolites between sample groups.

**Amino acid and derivative standards**

| | | |
|---|---|---|
| Creatine | Sigma | Cat#C0780 |
| L-Tryptophan | Sigma-Aldrich | Cat#T0254 |
| Creatinine | Sigma-Aldrich | Cat#C4255 |
| L-Citrulline | Sigma-Aldrich | Cat#C7629 |
| Sarcosine | Sigma-Aldrich | Cat#S7672 |
| L-Cysteine hydrochloride | Sigma | Cat#C1276 |
| L-Asparagine | Sigma | Cat#A0884 |
| Taurine | Sigma | Cat#T0625 |
| L-Ornithine monohydrochloride | Sigma-Aldrich | Cat#57197 |
| DL-2-Aminoadipic acid | Sigma | Cat#A0637 |
| Spermidine | Sigma-Aldrich | Cat#49761 |
| Putrescine | Sigma-Aldrich | Cat#51799 |
| L-Glutamine | Sigma-Aldrich | Cat#G3126 |
| Acetylcholine Choline | Sigma-Aldrich | Cat#A6625 |
| L-Alanine | National Institute of Metrology, China | GBW(E) 100062 |
| Glycine | National Institute of Metrology, China | GBW(E) 100062 |
| D-Serine | National Institute of Metrology, China | GBW(E) 100062 |
| L-Proline | National Institute of Metrology, China | GBW(E) 100062 |
| D-Valine | National Institute of Metrology, China | GBW(E) 100062 |
| L-Threonine | National Institute of Metrology, China | GBW(E) 100062 |
| L-Isoleucine | National Institute of Metrology, China | GBW(E) 100062 |
| L-Cystine | National Institute of Metrology, China | GBW(E) 100062 |
| L-leucine | National Institute of Metrology, China | GBW(E) 100062 |
| D-Lysine | National Institute of Metrology, China | GBW(E) 100062 |
| Glutamate | National Institute of Metrology, China | GBW(E) 100062 |
| DL-Methionine | National Institute of Metrology, China | GBW(E) 100062 |
| D-Arginine | National Institute of Metrology, China | GBW(E) 100062 |
| L-Tyrosine | National Institute of Metrology, China | GBW(E) 100062 |
| Aspartate | National Institute of Metrology, China | GBW(E) 100062 |
| L-Histidine | National Institute of Metrology, China | GBW(E) 100062 |
| Phenylalanine | National Institute of Metrology, China | GBW(E) 100062 |
| P5C | Jiangsu Aikang Biopharmaceutical | Custom made |

### Boxplot analysis

The Quartile (Quartile) calculation method was used to put the multiple sets of data drawing on the same coordinate, showing each group data of the distribution difference, and changes of the expression trends.

The targeted metabolomics data were deposited to the EMBL-EBI's MetaboLights repository (Haug et al, 2020) under accession code MTBLS7832.

## OCR and ECAR measurement using Seahorse cellular flux assays

Mitochondrial stress and glycolysis stress assays were performed using XF Cell Mito Stress Test kit (Agilent Technologies, #103015-100) and XF Glycolysis Stress Test Kit (Agilent Technologies, #103020-100), respectively. Naive and primed hESCs were planted at $8 \times 10^4$ or $6 \times 10^4$ cells per well on 96-well Seahorse plates that had been pre-coated with matrigel. After three days, culture media were exchanged for Seahorse XF DMEM Medium, pH 7.4 supplemented with 1 mM pyruvate, 2 mM glutamine, and 25 mM glucose (for the MitoStress assay), or 2 mM glutamine (for glucose stress assay) 1 h before the assay. Substrates and selective inhibitors were injected during the measurements to achieve final concentrations of oligomycin (1.5 μM), 4-(trifluoromethoxy) phenylhydrazone (FCCP, 0.5 μM), and rotenone/antimycin (Rot/AA, 0.5 μM) for OCR; glucose (10 mM), oligomycin (1 μM), and 2-Deoxy-d-glucose (2-DG, 50 mM) for ECAR. Metabolic flux analysis was performed using a Seahorse XFe 96 instrument. The OCR and ECAR values were normalized by the number of cells in each well.

## Measurement of ATP

An Enhanced ATP Assay Kit (Beyotime, #S0027) was used to assess cellular ATP content. Primed and naive (Rset and PXGL) hESCs were first lysed. After centrifugation, the supernatants were taken for determining the intracellular ATP levels. The detection solution was added to a 96-well plate and incubated at room temperature for 5 min. Consequently, the above mixed solutions were then added to the wells and mixed quickly before determining the luminescence signals using the Varioskan Flash spectral scanning multimode reader (Thermo Scientific). Total ATP levels were calculated based on the luminescence signals accordingly.

## Mitochondria and lysosomes imaging

Primed and naive hESCs, grown on glass bottom cell culture dishes, were incubated with NucBlue™ Live ReadyProbes™ Reagent (Invitrogen, #R37605) for 20 min at room temperature and then co-incubated with MitoTracker™ Green FM (Invitrogen, #M7514) and LysoTracker™ Deep Red (Invitrogen, #L12492) according to the manufacturer's instruction. Briefly, freshly prepared MitoTracker Green or LysoTracker Deep Red solution was added to medium to reach a final concentration of 150 nM and 50 nM, respectively. Cells were kept in the dark and maintained at 37 °C for 30 min, then washed with DMEM-F12 three times. After adding fresh culture medium, live cell images of mitochondria and lysosomes were captured using a Nikon A1R confocal microscope. In three separate trials, at least 30 cells were counted while photos were taken in a blinded manner. The length of mitochondria was assessed using the ImageJ software MiNA (Valente et al, 2017). Pearson's correlation coefficient (PCC) was calculated to quantitatively analyze the co-localization of mitochondria and lysosomes by Colocalization Finder, a plugin in ImageJ.

## ROS assay

Cellular ROS levels were determined by using the MitoSOX™ Red Mitochondrial Superoxide indicators (Thermo, #M36007) following the manufacturer's protocols. $4 \times 10^5$ of naive and primed hESCs were counted and collected. Then, cells were incubated at 37 °C for 30 min in DMEM-F12 containing 500 nM of MitoSOX that reacts with superoxide to generate red fluorescence. Afterwards, the fluorescence was measured by an ACEA NovoCyte™ flow cytometer.

## Apoptosis assay

The PE Annexin V Apoptosis Detection Kit (BD Biosciences, #559763) was used to detect apoptotic cells. $10^5$ of naive and primed hESCs were harvested and resuspended in a binding buffer. After adding 5 μl of PE Annexin V and 5 μl of 7-AAD, the buffer was gently vortexed and incubated for 15 min at RT (25 °C) in the dark. Stained cells were examined by an ACEA NovoCyteTM flow cytometer.

## Detection of autophagic flux

According to the manufacturer's instructions, autophagy was measured using the Cyto-ID® Autophagy detection kit 2.0 (Enzo, #ENZ-KIT175-0050). For confocal microscopy detection, hESCs were grown on glass bottom cell culture dishes. Cells were washed with PBS and incubated with detection reagent for 30 min at 37 °C in the dark. After being fixed with 4% formaldehyde for 20 min at 4 °C in the dark, hESCs were incubated with Hoechst for 10 min to stain the nucleus. Finally, cells were visualized using a Nikon A1R confocal microscope. For flow cytometry analysis, $5 \times 10^5$ cells were pelleted at 1000 rpm for 3 min and incubated with cyto-ID green reagent as described above. An ACEA NovoCyteTM flow cytometer was used to analyze the fluorescent signals.

## Transmission electron microscopy

Primed and naive (Rset and PXGL) hESCs were fixed in 2.5% glutaraldehyde overnight at 4 °C and immersed in 1% osmium tetroxide for 1 h. After being stained with 2% uranyl acetate, samples were then sequentially dehydrated with 50, 70, 90, and 100% ethanol for 15 min each and then in 100% acetone for 20 min twice. Afterwards, ultrathin sections were prepared on copper grids using a Leica UC7 microtome (Leica, Solms, Germany). Lead citrate and uranyl acetate were used for counterstaining. Mitophagosomes were detected using a Tecnai 10 TEM (Philips, Amsterdam, Netherlands) at 100 kV.

## mtDNA/nDNA measurement

To determine the ratio of mitochondrial and nuclear DNA (mtDNA/nDNA), genomic DNA was isolated using the Ezup Column Animal Genomic DNA Purification Kit (Sangon Biotech, #

B518251), and real-time qPCR was carried out as Human Mitochondrial DNA (mtDNA) Monitoring Primer Set (TaKaRa, #7246). Briefly, quantitative PCR reactions were assembled as follows: 0.8 µL of template DNA (5 ng/µL isolated DNA), 0.4 µL of mtDNA or nDNA target-specific primer pair mix (400 nM final concentration each), 5 µL SYBR Green PCR Master Mix, and 3.8 µL $H_2O$ in 1 well of the 96-well PCR plate. All qPCR experiments were performed in triplicate wells in three independent experiments. The copy number of mtDNA was calculated from the Ct values obtained for each target gene following the instructions.

## Protein degradation detection

hESCs were treated with 20 µg/ml cycloheximide (CHX) for 24 h to block synthesis of nascent proteins. To determine if Complexes I-IV degradation occurred via a proteasome-dependent pathway, hESCs were pretreated with 5 µM MG-132 for 2 h, followed by addition of CHX and further incubation of 24 h still in the presence of MG-132. Whole-cell lysates were immunoblotted with the indicated antibodies. The WB bands were quantitated using ImageJ software. The mean protein levels of Complex I-IV were normalized by tubulin. Protein degradation ratio = (protein level in cells without CHX treatment - protein level in cells treated with CHX)/ protein level in cells without CHX treatment.

## Statistical analyses

*P*-values were calculated by two-tailed unpaired Student's t-test and one-way ANOVA with Bonferroni's multiple comparisons test using GraphPad Prism 9. As stated in the figure legends, the data were presented as mean ±S.D. The differences were considered as statistically significant when *P*-values were less than 0.05. All differences were shown as $*P < 0.05$, $**P < 0.01$, $***P < 0.001$.

# Data availability

The untargeted and targeted metabolomics data were deposited to the EMBL-EBI's MetaboLights repository under accession codes MTBLS7840 and MTBLS7832, respectively. All source data have been uploaded to the BioImage database with the accession number of S-BIAD964. The download link is https://www.ebi.ac.uk/biostudies/bioimages/studies/S-BIAD964?key=b5ed12a6-c092-4cd6-8731-cf9c4e61bf0a.

# Peer review information

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

## Acknowledgements

We are very grateful to Prof. Junfeng Ji for providing insightful advice. We thank LC-Bio Technology Co., Ltd. (Hangzhou, China) and Applied Protein Technology (Shanghai, China) for bioinformatics support. We thank Shuangshuang Liu and Yueting Xing from the Core Facilities, Zhejiang University School of Medicine for their technical support. We thank Yinping Lv in the Center of Cryo-Electron Microscopy (CCEM), Zhejiang University for her technical assistance on Transmission Electron Microscopy. This work was supported by grants from the National Natural Science Foundation of China (grant numbers 32170739; 82173080; 22075243), the National Key Research and Development Program of China (grant numbers 2016YFA0100303; 2016YFA0101201), the Zhejiang Provincial Natural Science Foundation of China (LY23H160002), the Postdoctoral Program of Shaoxing People's Hospital (2022BSH001), and a grant from the Independent Task of the State Key Laboratory for Diagnosis and Treatment of Infectious Diseases, the First Affiliated Hospital, School of Medicine, Zhejiang University.

## Author contributions

**Cheng Chen**: Data curation; Formal analysis; Investigation; Visualization; Methodology; Writing—original draft; Writing—review and editing. **Qianyu Liu**: Data curation; Formal analysis; Validation; Investigation; Visualization; Writing—original draft. **Wenjie Chen**: Data curation; Formal analysis; Validation; Investigation; Visualization; Writing—original draft. **Zhiyuan Gong**: Software; Formal analysis; Validation; Investigation; Visualization. **Bo Kang**: Resources; Data curation; Formal analysis; Investigation; Visualization. **Meihua Sui**: Conceptualization; Resources; Supervision; Funding acquisition; Methodology; Writing—original draft; Project administration. **Liming Huang**: Conceptualization; Resources; Supervision; Funding acquisition; Methodology; Project administration. **Ying-Jie Wang**: Conceptualization; Resources; Supervision; Funding acquisition; Methodology; Writing—original draft; Project administration; Writing—review and editing.

## Disclosure and competing interests statement

The authors declare no competing interests.

# Expanded View Figures

**Figure EV1.  PRODH knockdown had little effect on primed hESCs.**

(**A**) Alkaline phosphatase staining of primed and naive (Rset and PXGL) hESCs after PRODH KD. Images show the AP+ colony number in randomly selected fields. Scale bars, 2.5 mm. (**B**) The protein levels of the key pluripotency factors OCT4, NANOG, and SOX2 in PLKO.1 and shPRODH hESCs were examined by WB. (**C**) Quantifications of the WB bands relative to α-tubulin and normalized to H9P PLKO.1 values were shown. (**D**) Rset- and PXGL-induced naive H9 hESCs were harvested. QRT-PCR determined the mRNA levels of pluripotency marker and naive pluripotency marker genes. (**E**) There was no significant change in pluripotency marker and primed pluripotency marker genes between H9P-PLKO.1 and H9P-shPRODH hESCs. (**F**) PRODH KD in Rset and PXGL naive H9 cells resulted in increased transcription of mesendoderm markers. Data information: The statistical significance was analyzed using unpaired two-tailed Student's t-test in panels. NS, not significant ($P > 0.05$). *$P < 0.05$, **$P < 0.01$, ***$P < 0.001$. The data shown were all from three independent biological replicates. Each data point represents an independent biological replicate. Data were presented as mean ± SD.

▶

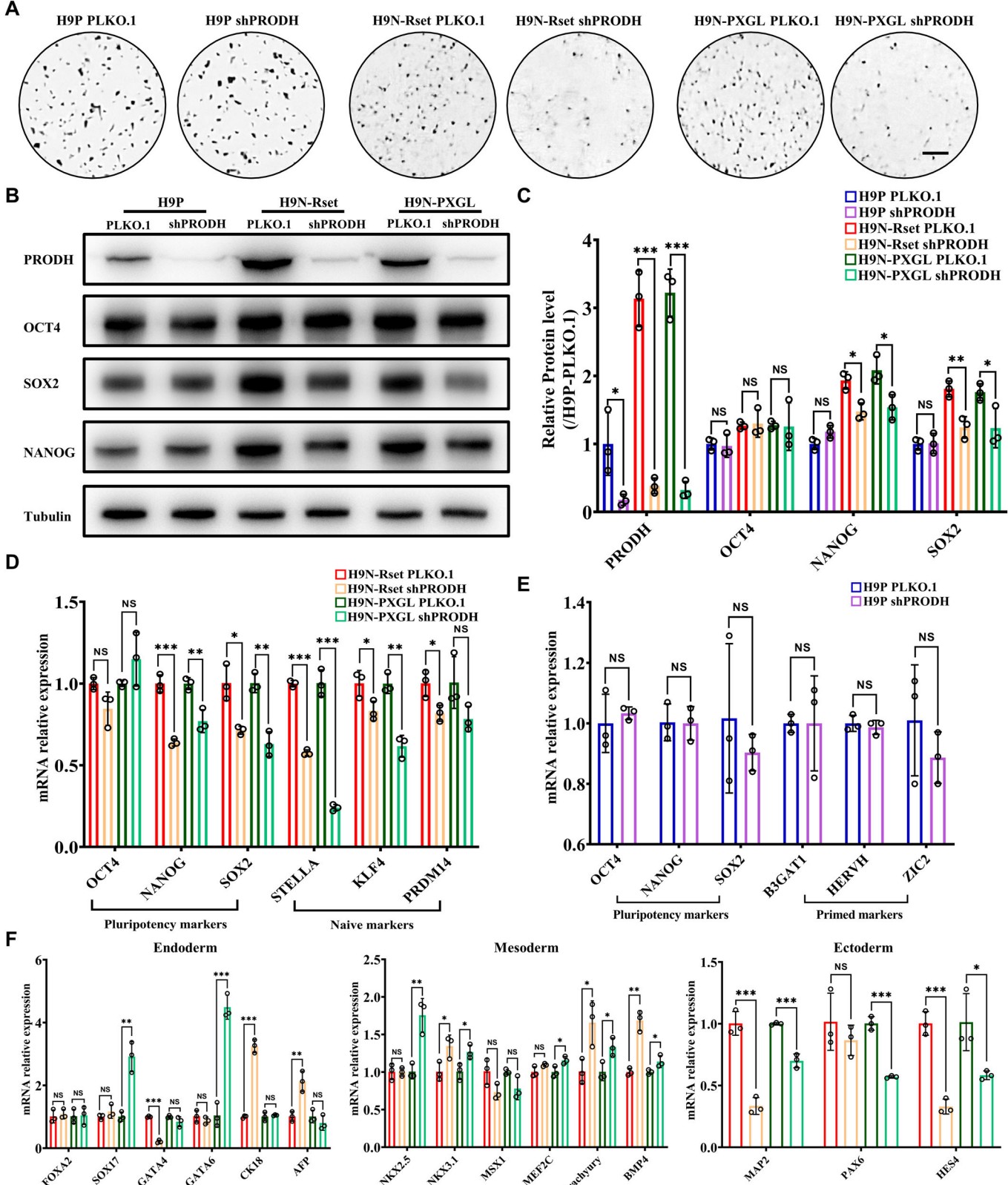

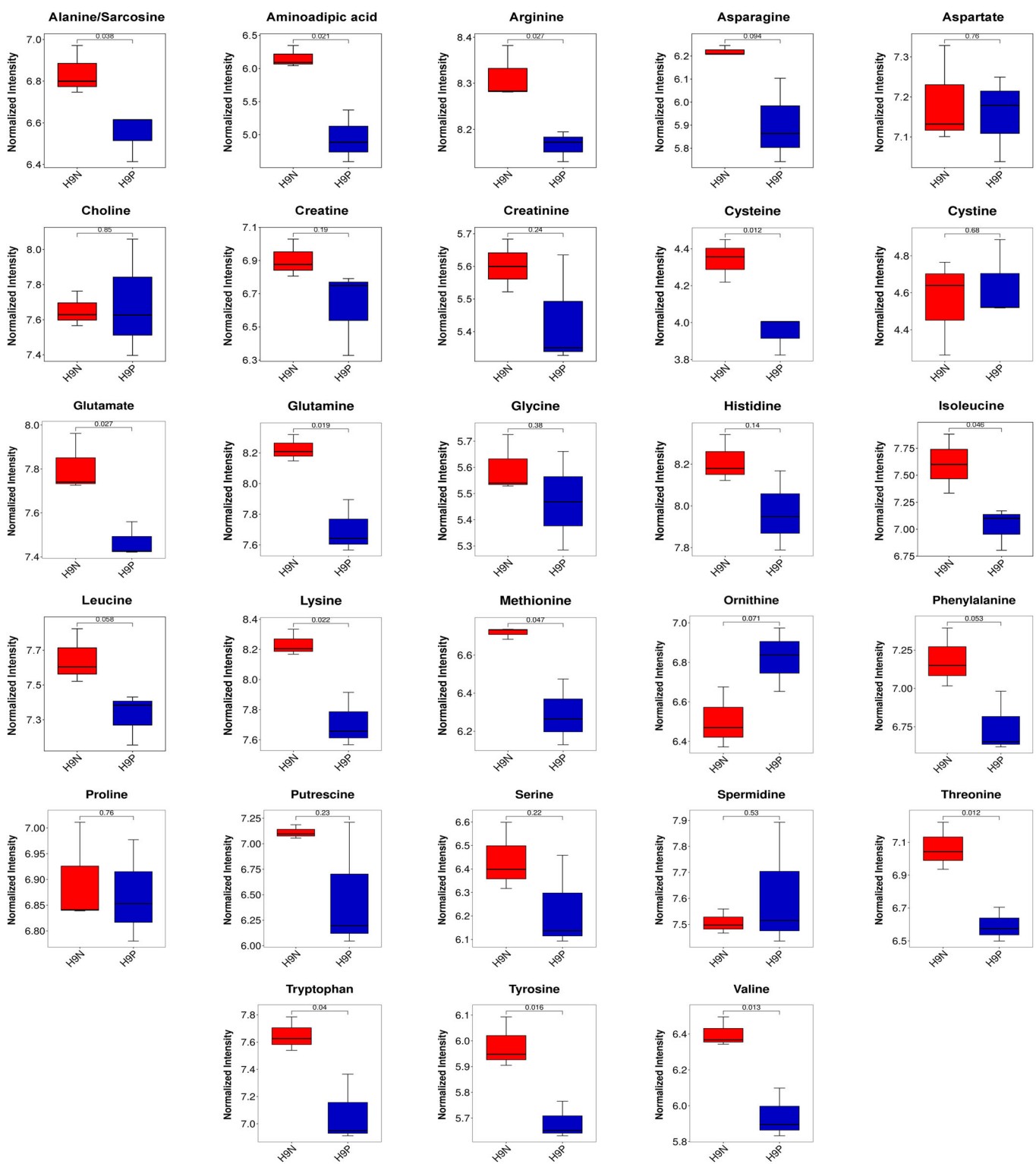

**Figure EV2. Analysis of targeted metabolomics of amino acids and their derivatives between naive (Rset) and primed H9 hESCs.**

The boxplots represent intensities of amino acids and their derivatives detected by targeted metabolomics. Normalized intensity, log10 (scale+1). Horizontal line represents median, box ranges represent values between quartiles 1 and 3, and whiskers represent values from minimum to maximum. Three biological replicates were analyzed. The statistical significance was analyzed using unpaired two-tailed Student's t-test.

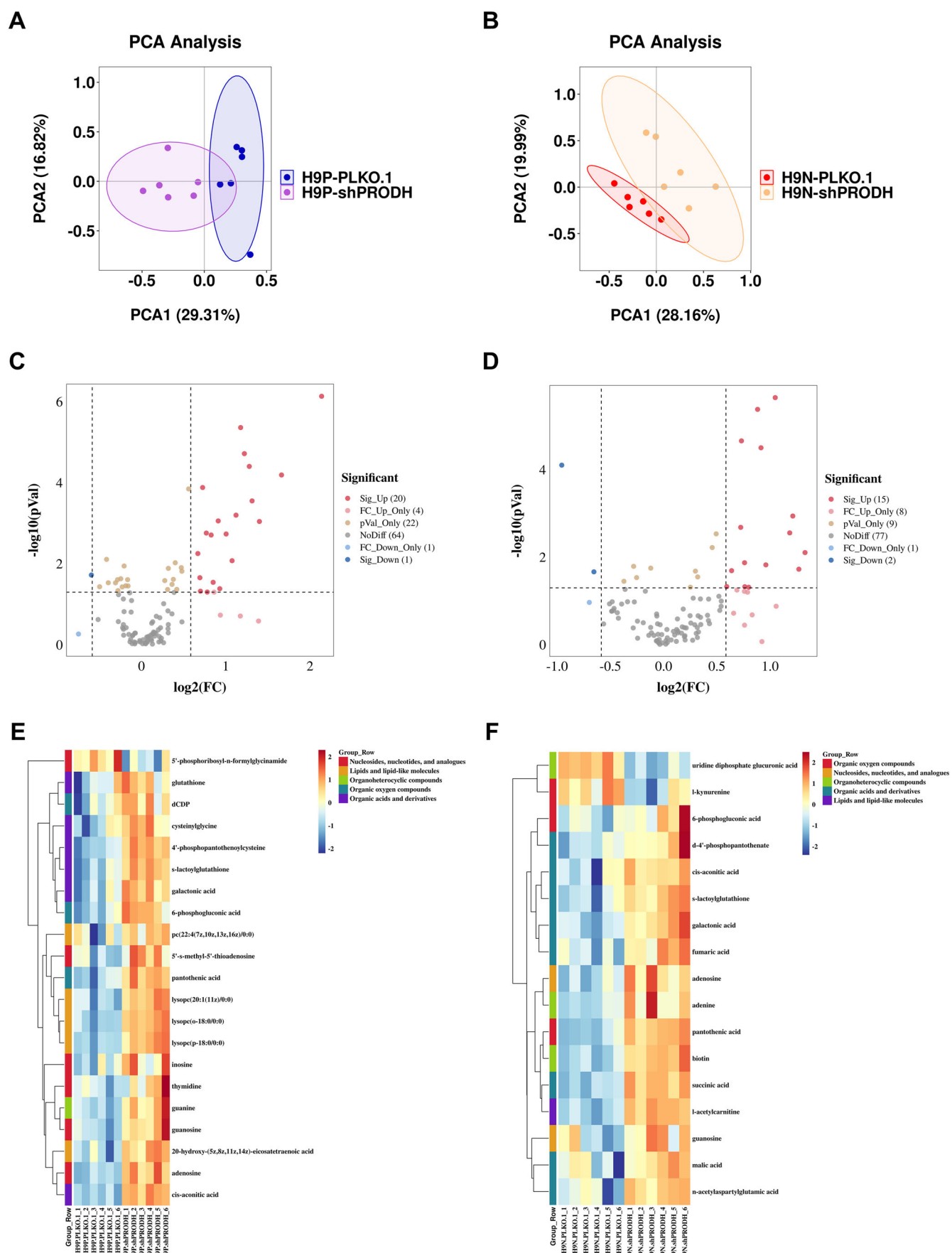

**Figure EV3.   The effect of PRODH knockdown on metabolism of primed versus naive (Rset) hESCs.**

(A,B) Principal-component analyses for all identified metabolites in H9P-shPRODH versus H9P-PLKO.1 hESCs (A) and H9N-shPRODH versus H9N-PLKO.1 hESCs (B). (C,D) Distribution of significantly changed metabolites (*P*-value < 0.05, FC > 1.5 or <0.67, as indicated by the dashed lines) between H9P-shPRODH and H9P-PLKO.1 hESCs (C), as well as H9N-shPRODH and H9N-PLKO.1 hESCs (D). The statistical significance was analyzed using unpaired two-tailed Student's t-test. (E,F) Heatmap showing significantly altered metabolites between H9P-shPRODH and H9P-PLKO.1 hESCs (E), as well as H9N-shPRODH and H9N-PLKO.1 hESCs (F).

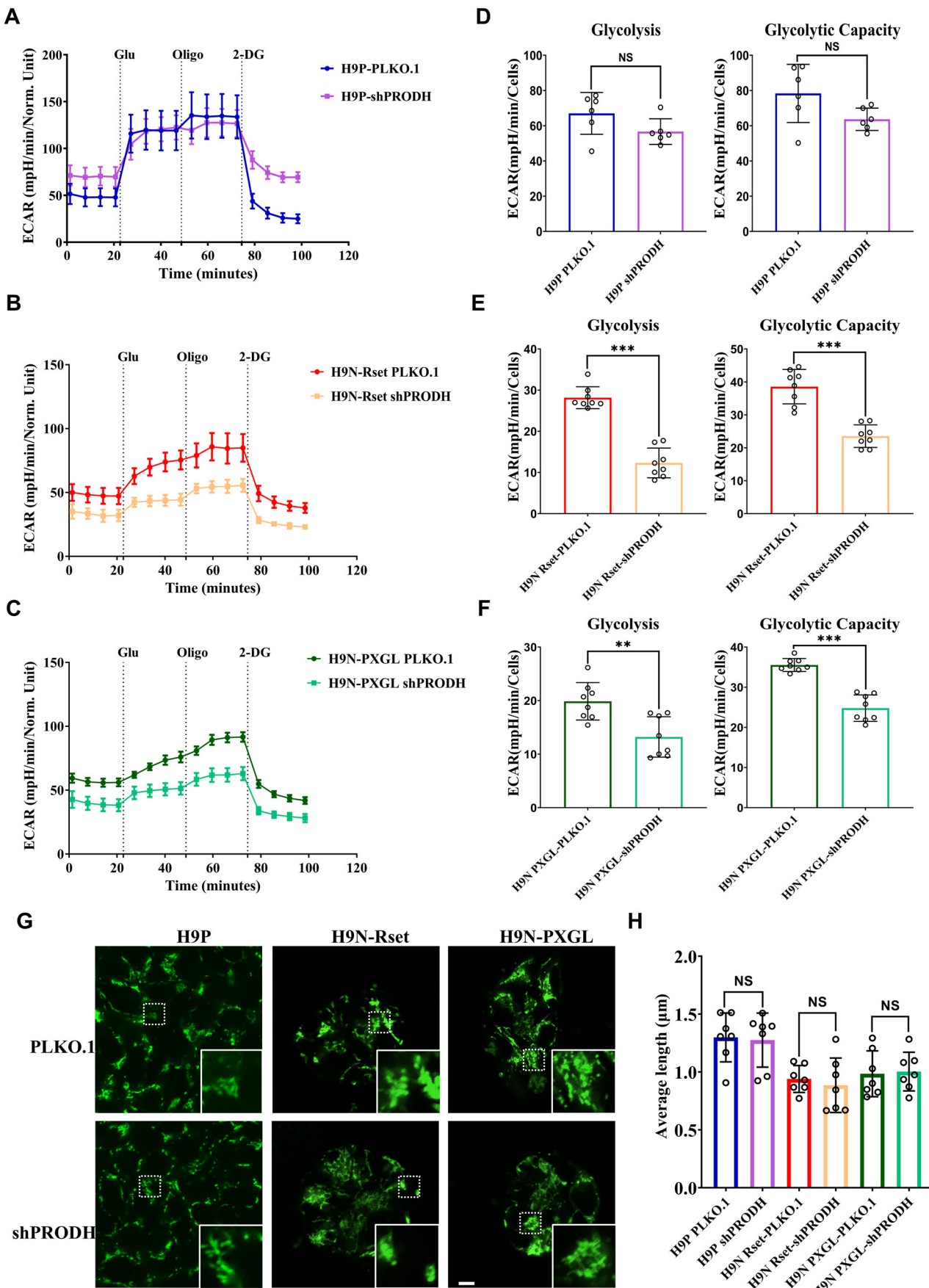

◀ **Figure EV4. Effect of PRODH knockdown on oxidative phosphorylation and glycolysis in primed versus naive hESCs.**

(A–C) Glycolytic function of PLKO.1 and shPRODH hESCs was measured by ECAR, after sequential injections of glucose (Glu), oligomycin (Oligo), and 2-deoxyglucose (2-DG). Ten technical replicates were analyzed. (D–F) ECAR of PLKO.1 and shPRODH hESCs in mTeSR1 (D), Rset (E), and PXGL (F) mediums were detected and analyzed. Glycolysis was the increase in ECAR measured after the glucose injection. Glycolytic capacity was defined by subtracting ECAR with 2-DG from ECAR with oligomycin. (G,H) The length of mitochondria. Mitochondria were stained with MitoTracker Green for 30 min. Representative confocal micrographs are shown (G). Scale bars, 10 μm. Mitochondrial length were analyzed by the ImageJ software MiNA (H). Data information: The statistical significance was analyzed using unpaired two-tailed Student's t-test in panels. NS, not significant ($P > 0.05$). **$P < 0.01$, ***$P < 0.001$. Data were presented as mean ± SD. In (D–F), each data point represents a technical replicate. In (G and H), data were obtained from seven independent experiments, and more than ten cells were analyzed in each experiment. Each data point represents the average mitochondrial length in each replicate.

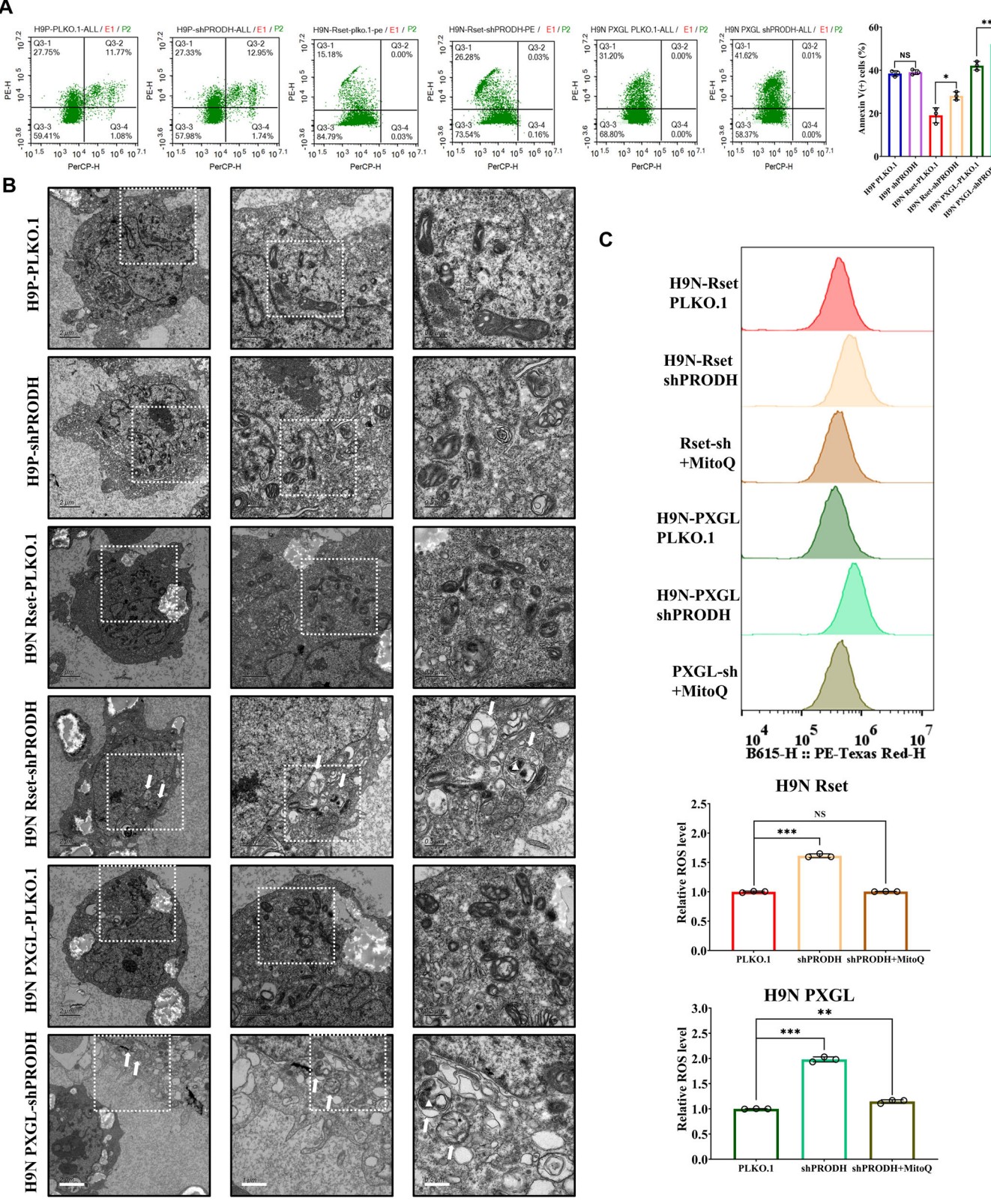

**Figure EV5.   The impact of excessive ROS on naive hESCs.**

(A) Cells stained with Annexin V and 7AAD were determined by flow cytometry. Live cells were defined as Annexin V-/7AAD-, early apoptosis AnnexinV + /7AAD-, and late apoptosis AnnexinV + /7AAD+. (B) Primed and naive (Rset and PXGL) hESCs were harvested for examination by transmission electron microscopy (TEM). The images were shown at various magnification levels. Scale bars, 2 μm (left), 1 μm (middle) 0.5 μm (right). The white arrows indicate AV-like organelles and the white triangles indicate mitochondria enclosed within AVs. (C) ROS levels were determined by MitoSOX. Rset-shPRODH and PXGL-shPRODH cells were treated with 50 nM MitoQ for 3 h, and cultured for an additional 12 h. Fluorescent intensity of cells stained with MitoSOX was determined by flow cytometry (upper), and quantitative data normalized to Control cell values (H9N Rset-PLKO.1 or H9N PXGL-PLKO.1) were presented (lower). Data information: The statistical significance was analyzed using unpaired two-tailed Student's t-test in panels. NS, not significant ($P > 0.05$). $^*P < 0.05$, $^{**}P < 0.01$, $^{***}P < 0.001$. The data shown were all from three independent biological replicates. Each data point represents an independent biological replicate. Data were presented as mean ± SD.

