## [Peer Review File · EMBO Reports]

PRODH safeguards human naïve pluripotency by limiting reactive oxygen species production

Cheng Chen, Qianyu Liu, Wenjie Chen, Zhiyuan Gong, Bo Kang, Meihua Sui, Liming Huang, and Ying-Jie Wang

Corresponding author(s): Ying-Jie Wang (yingjiawang@zju.edu.cn), Meihua Sui (suim@zju.edu.cn), Liming Huang (0622122@zju.edu.cn)

Review Timeline:

Submission Date:	3rd Jun 23
Editorial Decision:	24th Jul 23
Revision Received:	16th Dec 23
Editorial Decision:	2nd Feb 24
Revision Received:	8th Feb 24
Accepted:	20th Feb 24

Editor: Deniz Senyilmaz Tiebe

Transaction Report:

Dear Prof. Wang,

Thank you for the submission of your research manuscript to our journal, which was now seen by three referees, whose reports are copied below.

The referees express interest in the proposed role of PRODH in naïve pluripotency of hESCs. However, they also raise significant concerns that need to be addressed to consider publication here. In particular,

In particular, referees point out issues regarding

- the proposed link between MYC and PRODH1 (referee #1, point 2; referee #2, point 3)
- interpretation of metabolomics data (referee #3)
- data quantification and reproducibility (all referees).

Should you be able to address the referee concerns fully, we would like to invite you to submit a revised manuscript. Please revise your manuscript with the understanding that the referee concerns (as in their reports) must be fully addressed and their suggestions taken on board. Please address all referee concerns in a complete point-by-point response. Acceptance of the manuscript will depend on a positive outcome of a second round of review. It is EMBO reports policy to allow a single round of major experimental revision only and acceptance or rejection of the manuscript will therefore depend on the completeness of your responses included in the next, final version of the manuscript.

We realize that it is difficult to revise to a specific deadline. In the interest of protecting the conceptual advance provided by the work, we recommend a revision within 3 months. Please discuss the revision progress ahead of this time with me if you require more time to complete the revisions, or if you have questions or comments regarding the revision (also by video chat).

1. A data availability section providing access to data deposited in public databases is missing (where applicable).
2. Your manuscript contains statistics and error bars based on $n=2$. Please use scatter plots in these cases.

You can submit the revision either as a Scientific Report or as a Research Article. For Scientific Reports, the revised manuscript can contain up to 5 main figures and 5 Expanded View figures, and it should not exceed 27000 characters. If the revision leads to a manuscript with more than 5 main figures it will be published as a Research Article. In this case the Results and Discussion section should be separate. If a Scientific Report is submitted, these sections have to be combined. This will help to shorten the manuscript text by eliminating some redundancy that is inevitable when discussing the same experiments twice. In either case, all materials and methods should be included in the main manuscript file.

3) We replaced Supplementary Information with Expanded View (EV) Figures and Tables that are collapsible/expandable online. A maximum of 5 EV Figures can be typeset. EV Figures should be cited as 'Figure EV1, Figure EV2' etc... in the text and their respective legends should be included in the main text after the legends of regular figures.

4) a .docx formatted letter INCLUDING the reviewers' reports and your detailed point-by-point responses to their comments. As part of the EMBO publication's Transparent Editorial Process, EMBO reports publishes online a Review Process File (RPF) to

accompany accepted manuscripts. This File will be published in conjunction with your paper and will include the referee reports, your point-by-point response and all pertinent correspondence relating to the manuscript.

<https://www.embopress.org/page/journal/14693178/authorguide#transparentprocess>

5) a complete author checklist, which you can download from our author guidelines

<https://www.embopress.org/page/journal/14693178/authorguide>. Please insert information in the checklist that is also reflected in the manuscript. The completed author checklist will also be part of the RPF.

6) Please note that all corresponding authors are required to supply an ORCID ID for their name upon submission of a revised manuscript (<<https://orcid.org/>>). Please find instructions on how to link your ORCID ID to your account in our manuscript tracking system in our Author guidelines

<<https://www.embopress.org/page/journal/14693178/authorguide#authorshipguidelines>>

Additional information on source data and instruction on how to label the files are available:

<https://www.embopress.org/page/journal/14693178/authorguide#sourcedata>

9) Our journal encourages inclusion of *data citations in the reference list* to directly cite datasets that were re-used and obtained from public databases. Data citations in the article text are distinct from normal bibliographical citations and should directly link to the database records from which the data can be accessed. In the main text, data citations are formatted as follows: "Data ref: Smith et al, 2001" or "Data ref: NCBI Sequence Read Archive PRJNA342805, 2017". In the Reference list, data citations must be labeled with "[DATASET]". A data reference must provide the database name, accession number/identifiers and a resolvable link to the landing page from which the data can be accessed at the end of the reference. Further instructions are available at <http://www.embopress.org/page/journal/14693178/authorguide#referencesformat>

10) Regarding data quantification (see Figure Legends:

<https://www.embopress.org/page/journal/14693178/authorguide#figureformat>)

11) The journal requires a statement specifying whether or not authors have competing interests (defined as all potential or actual interests that could be perceived to influence the presentation or interpretation of an article). In case of competing

interests, this must be specified in your disclosure statement. Further information: <https://www.embopress.org/competing-interests>

12) Please also note our reference format:

I look forward to seeing a revised version of your manuscript when it is ready. Please let me know if you have questions or comments regarding the revision.

Kind regards,

Deniz Senyilmaz Tiebe

Deniz Senyilmaz Tiebe, PhD
Editor
EMBO Reports

Referee #1:

In this manuscript the authors analyse the role of PRODH1 in human embryonic stem cell (hESCs) pluripotency. They found higher PRODH1 expression in naïve hESCs compared to primed hESCs and that PRODH1 knock-down affects the proliferation and possibly also the pluripotent identity of naïve cells, but not of primed cells. They also observed that MYC appears to regulate PRODH1 expression, as MYC inhibition increased PRODH1 levels and its inhibition appears to decrease MYC expression when cells are treated with the ERK1/2 inhibitor PD0325901. Surprisingly, the authors found no difference in proline levels between naïve and primed cells and only a trend for higher proline levels after PRODH1 knockdown. Additionally, exogenous proline did not affect naïve hESCs. Interestingly, however, PRODH1 knockdown increases respiration, induces higher ROS, higher levels of DNA damage and increases mitophagy in naïve but not primed hESCs. These observations also correlate with a decrease in mitochondrial DNA and OXPHOS proteins that the authors observe after PRODH1 knock-down. Finally, the authors found that inhibiting translation caused a slower degradation of OXPHOS components in PRODH1 knock-down cells than in control cells, suggesting that degradation was slower in these cells, possibly due to the mitophagy differences.

This is an interesting paper, but that would benefit from some changes to make the data more robust.

The primary issue relates to the lack of quantification. The western blots and immunofluorescence experiments need to be quantified to provide insight into how reproducible the data is. This lack of quantification also impacts 3 conclusions of the manuscript, that may be better supported if quantification is provided.

1. The role of PRODH1 in regulating pluripotent gene expression is unclear. The experiments/quantification provided do not allow to distinguish this potential role from its role in sustaining proliferation.
2. The interaction between MYC and PRODH1 is unclear. It is unclear why MYC over-expression decreases PRODH1 expression only when cells are treated with PD0325901. Also, in this condition MYC levels seem to be decreased rather than increased, so how do the authors know that this effect is due to MYC?
3. The lack of quantification leads to the data shown not supporting the conclusion that inhibiting ROS levels with MitoQ restores pluripotency factor expression or proliferation.

Also, the conclusion that PRODH1 does not regulate proline levels could be better supported by the data. For example, the significance of the differences in proline levels are unclear, as exogenous proline is added via the media that naïve and primed cells are cultured in. Do naïve and primed cells see similar levels of exogenous proline? The authors may need to normalise proline levels between conditions or perform their experiments in proline free media to make their conclusions more robust. The first will be easier to do than the second. At the bare minimum they should discuss how exogenous proline may affect their conclusions. Similarly, in the experiments in which 2.5mM exogenous proline is provided, how much does this change the overall levels of proline provided in the culture media?

Referee #2:

The manuscript of Chen et al., entitled "PRODH safeguards human naïve pluripotency by limiting mitochondrial oxidative phosphorylation and reactive oxygen species production" reports on the role of proline dehydrogenase (PRODH) in hESCs. The author found that in naïve hESC, PRODH gene expression is repressed by c-Myc, which in turn is activated by the MEK inhibitor (PD) present in the culture medium. Using well-planned experiments, the authors found that the downregulation of PRODH in

naive hESC boost OXPPOS and ROS production, triggering apoptosis via DNA damage and autophagy. These effects can be to a large extent rescued by a mitochondria-targeted antioxidant. The authors investigated the proline metabolism and the effects of PRODH k.d. on ETC components. Altogether, the authors concluded that PRODH function in naïve hESC plays a critical role in tuning down OXPPOS and ROS production, thereby safeguarding the hESCs self-renewal.

The manuscript is well-written and the experimental result generally supports the authors' main conclusions. I have several suggestions and questions as follows.

1. In the abstract, the authors state that "Thus, we revealed a crucial role of PRODH in limiting mtOXPHOS and ROS production, and thereby safeguarding naïve pluripotency of hESCs during early embryonic development." There are no experiments that examine the PRODH function during embryonic development. All experiments are performed in hESCs. I suggest removing "early embryonic development" or providing embryo data.
2. The authors showed the naive hESCs differentiated toward mesendoderm upon deletion of PRODH. I am wondering whether these cells directly differentiate into mesendoderm cells or through the primed state to the mesendoderm state. Taking into account that PRODH does not affect the primed cells, what could be the explanation for the differentiation into this lineage? This can be discussed.
3. In Fig 2J, why does the level of Myc in the PD + OE-MYC condition appear even lower than in the PD-treated condition? If indeed MEK affects cMyc expression via miR-23b how is it possible that PD affects the expression of the ectopic cMyc? Can miR-23b bind the ectopic cMyc sequences? Are there any other possibilities linking MEK activity and cMyc expression apart from miR-23b?
4. To establish the correlation between ROS and components of ETC, I am wondering whether MG132-treated cell with a high level of ETC proteins shows a high level of ROS.
5. Figure 5F, Nanog shows 2 bands in H9N-Rset and one in H9N-PXGL.
6. Figure 6E and F, CIII-R band, the shape looks strange.
7. Myc shows one band in Fig.2H and two bands in Fig.2J
8. How is Myc expression in the shPRODH group compared to the control group (in both naïve and primed cells)?
9. When limiting ROS production by MitoQ treatment, are there any effects on mtOXPHOS and glycolysis (feedback loops)? This is not a critical question and it is up to the authors to decide whether they would like to address it experimentally.
10. PRODH k.d. show no effect on proline metabolism. Why? Are there any compensatory factors? This can be discussed.
11. Can the authors provide any statistics on EV6D (ROS levels)?

Referee #3:

The manuscript from Chen and colleagues describes a study to investigate the role of PRODH - a mitochondrial enzyme that oxidises proline - in embryonic stem cell phenotype. While it has previously been shown that PRODH expression is regulated by MYC, it has not been described in ESCs, or through a culture supplement (the MEK inhibitor, PD0325901). While this is a potentially exciting story, the issues of data quality throughout mean that it is difficult to be convinced that the narrative is correct. There are some exciting experiments in there that really suggest something is going on, but overall the data quality and experimental design needs to be improved. Particular concern needs to be taken over the mass spectrometry metabolomic data, which is problematic (as outlined below). Overall the style of the manuscript makes it impenetrable for a non-expert of the ESC field, particularly at the beginning. The language needs to be decoded throughout to make it understandable for more than a niche audience.

The authors note that knockdown of PRODH decreased AP+ colony formation (Figure 1E). Did it reduce colony number overall? It appears to have reduced colony size from the image. They also in the same section note that it reduced NANOG expression in the PXGL-cultured ESCs - did it also change colony size and AP+ expression, given that these are both models of naïve-like ESCs? What about NANOG expression in the Rset culture? Uncertain, given evidence in Ex Figure 1A whether this effect is only in a single model (PXGL) as the loading is uneven in the Rset blot, and the reduction in NANOG may not be significant. Indeed in H9P primed cells, all three markers may increase in expression. The images shown in Ex Figure 1E are highly unclear, alongside that of Figure 1E - these need to be improved to be informative.

Reproducibility of the data are unclear throughout - it is not apparent how many times any experiment was performed except in a minority of cases, whether replicates were taken within each experiment, etc.

In describing their metabolomic data (EV2A and B) the authors suggest that as there are more upregulated than downregulated metabolites, there is 'more active metabolic activity' within naïve hESCs. This is not an accurate interpretation of these data - decreases in metabolites can also indicate pathway activity (due to a higher rate of use) and increased metabolites can indicate blocked pathways. They should reconsider this description. The list of metabolites appears to be via putative ID, and therefore there are a large number of metabolites that should not be possible to observe in mammalian cell culture, including ibulast (a drug), bakers yeast extract, vinyl acetic acid, clidinium (another drug), etc. Further examples of this lie in the EV4E plot, showing compounds such as rhodamine b, suprofen and penicillin g varying between the two groups. It significantly reduces confidence in the expertise used to analyse the dataset, and the identity of any of the metabolites observed and reported. Given this, and the reported difficulty in detecting P5C in cell extracts in the field, the result shown in Figure 3A (P5C in H9N cells) needs to be confirmed with a chemical standard.

Figure 3E shows that proline does not change between the comparisons (control versus shPRODH) and also does not appear to figure in the heatmap in EV data of those metabolites that changed. This strongly suggests that proline metabolism is not involved in the changes observed with shPRODH, thereby undermining the manuscript's conclusions. It therefore makes no logical sense to pursue this further as a line of investigation. The enrichment of citric acid cycle metabolites shown (Figure 4A), which reflect a representation of the heatmap (EV4) also does not appear to link - other than succinate, there is no other change in TCA cycle metabolites that is present in the heatmap, and it cannot be that one can infer a change in a whole metabolic pathway based on the changes in one metabolite that can be made through a number of other routes (e.g. 2-oxoglutarate-dependent dioxygenase activity). That being said, there is clearly a significant change in respiration in the shPRODH H9N cells (Rset condition) - more than doubling of the rate. Given that glycolysis shows around 20% decrease, these analyses suggest strongly that it is something other than ATP generation that is driving this change - therefore the authors conclusions of this section (pdf page 7) are not supported. The energy generation change by 2x OXPHOS is around 36X ATP if the coupling of the respiratory chain doesn't change. This is clearly not balanced by a 20% reduction in ATP generation by glycolysis. While not as extreme, a similar change is seen for the H9N PXGL state.

It is incorrect to say that 'mtOXPHOS [is] an energy metabolism that happens to be substantially weakened in the primed state'. Scientific terminology is important here, and referencing of statements such as these to support this type of remark (top of page 7 of the pdf).

The ROS kit used to measure these species is based on DCFDA fluorescence, which mainly picks up cellular peroxide, and is not mitochondrial specific. The authors do not present data to support mitochondrial ROS production.

Co-localisation of mitochondria and lysosomes shown in Extended Figure 6C needs to be at higher resolution to be convincing - as the mitochondria cover most of the non-nuclear intracellular space, it is difficult to know for sure whether the colocalization is due to resolution rather than biology.

The reviewers' comments are very pertinent and constructive, and we would like to address them point-by-point as follows:

Reviewer #1: In this manuscript the authors analyse the role of PRODH1 in human embryonic stem cell (hESCs) pluripotency. They found higher PRODH1 expression in naïve hESCs compared to primed hESCs and that PRODH1 knock-down affects the proliferation and possibly also the pluripotent identity of naïve cells, but not of primed cells. They also observed that MYC appears to regulate PRODH1 expression, as MYC inhibition increased PRODH1 levels and its inhibition appears to decrease MYC expression when cells are treated with the ERK1/2 inhibitor PD0325901. Surprisingly, the authors found no difference in proline levels between naïve and primed cells and only a trend for higher proline levels after PRODH1 knock-down. Additionally, exogenous proline did not affect naïve hESCs. Interestingly, however, PRODH1 knock-down increases respiration, induces higher ROS, higher levels of DNA damage and increases mitophagy in naïve but not primed hESCs. These observations also correlate with a decrease in mitochondrial DNA and OXPHOS proteins that the authors observe after PRODH1 knock-down. Finally, the authors found that inhibiting translation caused a slower degradation of OXPHOS components in PRODH1 knock-down cells than in control cells, suggesting that degradation was slower in these cells, possibly due to the mitophagy differences.

Comments:

This is an interesting paper, but that would benefit from some changes to make the data more robust. The primary issue relates to the lack of quantification. The western blots and immunofluorescence experiments need to be quantified to provide insight into how reproducible the data is.

Response: We appreciate the reviewer's precise summary and valuable advice. Following the reviewer's suggestion, we performed rigorous quantitative analysis and statistical significance test for all immunoblotting and immunofluorescence data as well as other quantifiable data, and incorporated them into the revised manuscript, presented in revised Fig 1B/E/F/H, Fig 2B-J, Fig 3C/F/G, Fig 5A/C/E-G, Fig 6, Fig EV1B-F, Fig EV4, Fig EV5A/C, Appendix Fig2, and Appendix Fig3. We specified biological or technical replicate numbers and the *P* values of statistical analyses in the above-mentioned figures and their legends. In addition, we also uploaded all source data (including the replicate data). These efforts made our data more robust and conclusions more convincing.

This lack of quantification also impacts 3 conclusions of the manuscript, that may be better supported if quantification is provided.

1. The role of PRODH1 in regulating pluripotent gene expression is unclear. The experiments/quantification provided do not allow to distinguish this potential role from its role in sustaining proliferation.

Response: We thank the reviewer for this critical point. A total of four sets of experiments were employed to assess the naïve versus primed pluripotency of hESCs: AP staining, Immunofluorescence, Western blotting, and RT-qPCR.

For AP staining, we replaced the dye, conducted a new set of experiments, and quantified the results. The blue dye manifested a pronounced reduction in the number of AP-positive colonies derived from naïve hESCs following the knock-down of PRODH (revised Figs 1E and EV1A). For immunofluorescence microscopy, we repeated the experiments using hESCs treated and cultured under six conditions. Notably, between naïve hESC clumps with comparable size and morphology, shPRODH-transfected cells had significantly diminished NANOG immunofluorescence signals compared to PLKO.1-transfected control cells (revised Figs 1F). Additionally, we repeated Western blot experiments, comparing hESCs treated and cultured under six conditions on the same gel and quantified the results. Tubulin was served as a sample loading control to normalize the differences in total cell/protein numbers of different samples loaded onto the gel. The levels of NANOG and SOX2 in Rset/PXGL-shPRODH cells were approximately 70~80% of those of control group (revised Figs EV1B and EV1C). When integrated with qPCR data, our findings revealed that after PRODH knock-down, the transcript levels of NANOG and SOX2 decreased, mesendoderm marker genes increased and ectoderm marker genes decreased in naïve hESCs, and the cells exhibited differentiation morphology towards the mesendoderm (revised Figs EV1D-F). All experiments above were conducted in three independent biological replicates, and the raw data have been uploaded.

Taken together, the above comprehensive analyses led us to the conclusion that PRODH genuinely regulates pluripotency in the naïve state, dispelling concerns of its impact being an artifact induced by inhibiting cell proliferation.

2. The interaction between MYC and PRODH1 is unclear. It is unclear why MYC over-expression decreases PRODH1 expression only when cells are treated PD0325901. Also, in this condition MYC levels seem to be decreased rather than increased, so how do the authors know that this effect is due to MYC?

Response: We appreciate the comment. *It has been demonstrated in other cellular systems that MYC suppressed POX/PRODH expression primarily through upregulating miR-23b*, which selectively reduces translation by binding to the 3'-untranslated region of the mRNA PRODH/POX with a particular sequence (Liu et al, 2012), and the expression level of Myc is regulated by MEK/ERK pathway: ERK phosphorylates Ser62 and stabilizes Myc (Farrell & Sears, 2014).* As illustrated in the revised Figs 2A-E, the key factor in culture media that drove the differential PRODH expression between primed and naïve states was identified as PD0325901, a MEK/ERK inhibitor. Therefore, we examined three key regulatory elements upstream of PRODH—P53, AMPK, and MYC—guided by existing literature, and found PD0325901 reduced the protein level of MYC but did not affect that of P53 and AMPK (revised Fig 2F). Besides, both Rset and PXGL naïve hESCs, as well as primed H9 cells treated with PD0325901, showed a reduction in MYC mRNA levels, consistent with the high level of PRODH in these hESCs (revised Fig 2G). Moreover, MYC knockdown enhanced transcript and protein expression levels of PRODH in primed H9 hESCs, partially phenocopying PD0325901 treatment (revised Fig 2H and 2I).

To further explore the interaction between MYC and PRODH, we overexpressed ectopic MYC in primed H9 (H9P) cells supplemented with or without PD0325901. *Yet, the variable plasmid transfection efficiency inherent in hESCs resulted in fluctuations in MYC expression levels that may artificially led to a decreased MYC band in the last lane of the original Fig 2J. To*

overcome this, we used lentivirus to establish cell lines with stably overexpressed MYC and repeated the WB experiment (revised Fig 2J). While overexpressing MYC did not lead to a further reduction in PRODH level, indicating a potential limit of MYC's inhibitory effect on PRODH (H9P OE-MYC vs. H9P), it did, however, partially impede the elevation of PRODH level induced by PD0325901 (H9P OE-MYC+PD vs. H9P+PD). Taken together, we reveal that PD0325901 critically upregulated PRODH protein levels in hESCs by suppressing MYC. We summarized the MEK/ERK-MYC-PRODH signaling pathway in Appendix Fig2A, and added more description into the revised manuscript. All experiments above were conducted in three independent biological replicates and quantified. Raw data were uploaded.

3. The lack of quantification leads to the data shown not supporting the conclusion that inhibiting ROS levels with MitoQ restores pluripotency factor expression or proliferation.

Response: Following the reviewer's advice, we quantified the WB data in Fig 5F. We also detected the fluorescence intensity of EdU using flow cytometry and further quantified the results. Both sets of results consistently demonstrated that the inhibition of ROS with MitoQ effectively restored the compromised pluripotency and proliferation capabilities observed in shPRODH cells in naïve state (revised Figs 5F and G). All experiments above were conducted with three independent biological replicates, and raw data has been uploaded. We showed biological replicates as data points in the indicated figures.

Also, the conclusion that PRODH1 does not regulate proline levels could be better supported by the data. For example, the significance of the differences in proline levels are unclear, as exogenous proline is added via the media that naïve and primed cells are cultured in. Do naïve and primed cells see similar levels of exogenous proline? The authors may need to normalise proline levels between conditions or perform their experiments in proline free media to make their conclusions more robust. The first will be easier to do than the second. At the bare minimum they should discuss how exogenous proline may affect their conclusions.

Response: We are grateful for the reviewer's pertinent and constructive comments. *Indeed, the similar proline content in the Rset (0.2mM) vs mTeSR1 (0.216mM) culture systems led us to select this pair culture condition, rather than the PXGL (0.11mM) vs mTeSR1 (0.216mM) pair condition, for both targeted and untargeted metabolomics assays.* We added a detailed description into the revised manuscript.

From the available published literature, we ascertained that *the proline content in the mTeSR1 culture system is 0.216mM* (Ludwig *et al*, 2006). To find out the proline content in the Rset and PXGL systems, we contacted the manufacturer STEMCELL Technologies who responded that L-Proline would be in the range noted in the published formulations on which these products are based. Following this, we compiled a list of the additional components in the two culture systems based on relevant literature (Bredenkamp *et al*, 2019; Gafni *et al*, 2013) (Figs 2A).

Rset-feeder free: DMEM AlbuMAX(lipid-rich BSA) N2 supplement insulin LIF bFGF TGF β 1 L-glutamine NEAA β -mercaptoethanol	PXGL: N2B27 medium: DME/F12 Neurobasal B27 supplement N2 supplement L-glutamine β -mercaptoethanol	
	small molecule inhibitor: PD0325901(MEKi) CHIR99021(GSK3 β i) SP600125(JNKi) SB203580(p38i) Go6983(PKCi)	PDL/HDACi: PD0325901(MEKi) LIF VPA(HDACi)

Evidently, exogenous proline originates solely from the basal medium of the culture system (DMEM, DME/F12, Neurobasal) and the Non-Essential Amino Acid (NEAA) additives. The proline content within each component is presented according to Thermo's published formula: DMEM (Thermo, #12491015)-0.1mM; DME/F12 (Thermo, #12634010) -0.15mM; Neurobasal (Thermo, #21103049)-0.067mM; NEAA 100X (Thermo, #11140050)-10mM. Therefore, *for Rset, the proline content is calculated as $0.1+10/100=0.2\text{mM}$. For PXGL, the proline content is calculated as $(0.15+0.067)/2=0.11\text{mM}$.*

Therefore, *we chose Rset (0.2mM) and mTeSR1 (0.216mM) culture systems for both targeted and untargeted metabolomics assays.* In fact, *in all experimental systems where PRODH was knocked down (e.g., H9P-PLKO.1 vs. H9P-shPRODH; Rset-PLKO.1 vs. Rset-shPRODH; PXGL-PLKO.1 vs. PXGL-shPRODH), the PLKO.1- and shPRODH-transfected cells for comparison were cultured in the same culture media (either both in mTeSR1, Rset or PXGL media) that have identical exogenous proline content.*

Similarly, in the experiments in which 2.5mM exogenous proline is provided, how much does this change the overall levels of proline provided in the culture media?

Response: For Rset: $2.5\text{mM} / 0.2\text{mM} = 12.5$ fold; For PXGL: $2.5\text{mM} / 0.11\text{mM} = 22.7$ fold. Therefore, the supplemented 2.5mM exogenous proline is considered as largely excessive over the proline content in both culture systems.

Reviewer #2: The manuscript of Chen et al., entitled "PRODH safeguards human naïve pluripotency by limiting mitochondrial oxidative phosphorylation and reactive oxygen species production" reports on the role of proline dehydrogenase (PRODH) in hESCs. The author found that in naïve hESC, PRODH gene expression is repressed by c-Myc, which in turn is activated by the MEK inhibitor (PD) present in the culture medium. Using well-planned experiments, the authors found that the downregulation of PRODH in naïve hESC boost OXPHOS and ROS production, triggering apoptosis via DNA damage and autophagy. These effects can be to a large extent rescued by a mitochondria-targeted antioxidant. The authors investigated the proline metabolism and the effects of PRODH k.d. on ETC components. Altogether, the authors concluded that PRODH function in naïve hESC plays a critical role in tuning down OXPHOS and

ROS production, thereby safeguarding the hESCs self-renewal.

Comments:

The manuscript is well-written and the experimental result generally supports the authors' main conclusions. I have several suggestions and questions as follows.

1. In the abstract, the authors state that "Thus, we revealed a crucial role of PRODH in limiting mtOXPHOS and ROS production, and thereby safeguarding naïve pluripotency of hESCs during early embryonic development."

There are no experiments that examine the PRODH function during embryonic development. All experiments are performed in hESCs. I suggest removing "early embryonic development" or providing embryo data.

Response: We appreciate the reviewer's pertinent advice, and removed "early embryonic development" in the manuscript.

2. The authors showed the naïve hESCs differentiated toward mesendoderm upon deletion of PRODH. I am wondering whether these cells directly differentiate into mesendoderm cells or through the primed state to the mesendoderm state. Taking into account that PRODH does not affect the primed cells, what could be the explanation for the differentiation into this lineage? This can be discussed.

Response: This is a very interesting and stimulating question that we don't have a definitive answer. By using RNA-Seq profiling during specification to the three germ layers, Hanna and Pierre showed that mESCs (naïve state) switched on condition-specific gene expression programs from the onset of the differentiation procedure and that primed pluripotency did not constitute an obligatory intermediate state (Sladitschek & Neveu, 2019). Based on their finding, we tend to postulate that upon PRODH knock-down, naïve hESCs may directly differentiate into mesendoderm cells rather than going through the primed state. Further experiments are warranted to find out the answer.

3. In Fig 2J, why does the level of Myc in the PD + OE-MYC condition appear even lower than in the PD-treated condition?

If indeed MEK affects cMyc expression via miR-23b how it is possible that PD affects the expression of the ectopic cMyc? Can miR-23b bind the ectopic cMyc sequences? Are there any other possibilities linking MEK activity and cMyc expression apart from miR-23b?

Response: We appreciate the reviewer's questions. First, *the variable plasmid transfection efficiency inherent in hESCs resulted in fluctuations in MYC expression levels that may artificially led to a decreased MYC band in the last lane of the original Fig 2J. To overcome this, we used lentivirus to establish cell lines with stably overexpressed MYC and repeated the WB experiment (revised Fig 2J)*. While overexpressing MYC did not lead to a further reduction in PRODH level, indicating a potential limit of MYC's inhibitory effect on PRODH (H9P OE-MYC vs. H9P), it did, however, partially impede the elevation of PRODH level induced by PD0325901 (H9P OE-MYC+PD vs. H9P+PD). Taken together, we reveal that PD0325901 critically upregulated PRODH protein levels in hESCs by suppressing MYC.

Second, we would like to clarify, as shown in Appendix Fig 2A, that our hypothesis was miR-

23b* regulates PRODH expression but not MYC expression. *It has been demonstrated in other cellular systems that MYC suppressed POX/PRODH expression primarily through upregulating miR-23b*, which selectively reduces translation by binding to the 3'-untranslated region of the mRNA PRODH/POX with a particular sequence (Liu et al., 2012), and the expression level of Myc is regulated by MEK/ERK pathway: ERK phosphorylates Ser62 and stabilizes Myc (Farrell & Sears, 2014). We summarized the MEK/ERK-MYC-PRODH signaling pathway in Appendix Fig2A.*

4. To establish the correlation between ROS and components of ETC, I am wondering whether MG132-treated cell with a high level of ETC proteins shows a high level of ROS.

Response: We thank the reviewer for this insightful question. To answer this question, Rset-PLKO.1 and PXGL-PLKO.1 hESCs were incubated with 5 μ M MG-132 for 24 hours. ROS levels of cells stained with MitoSOX (superoxide indicators specifically targeted to mitochondria in live cells) were determined by flow cytometry, and quantitative data normalized to Control cell values (H9N Rset-PLKO.1 or H9N PXGL-PLKO.1) were presented (revised Fig 6D). MG132-treated cells with a high level of ETC proteins did exhibit higher levels of ROS, indicating a strong positive correlation between the components of ETC and ROS. The experiments were conducted in three independent biological replicates and quantified. These new results were added into the revised manuscript.

5. Figure 5F, Nanog shows 2 bands in H9N-Rset and one in H9N-PXGL.

Response: We thank the reviewer for careful examination of our data. This discrepancy is likely due to variations in gel concentration and electrophoresis duration, as the samples for Rset and PXGL were run on two separate gels/immunoblots in our first submission. We repeated the experiment and have now replaced the old blots with the new one where all samples were run on the same gel (revised Fig 5F). This set of experiment was conducted in three independent biological replicates and quantified, and the raw data were uploaded.

6. Figure 6E and F, CIII-R band, the shape looks strange.

Response: This may be because the bottom end of the protein gel did not solidify adequately. The bands of small molecular-weight proteins such as CIII-R will often be distorted when electrophoresed fully down to the bottom end of the gel. We re-run the samples with shorter time and replaced the old blot with a new one that shows CIII-R bands with normal shape.

7. Myc shows one band in Fig.2H and two bands in Fig.2J.

Response: Again, we thank the reviewer for careful examination of our data. In fact, c-Myc showed two bands in most of our experiments and figures (Fig 2F, 2H and 2J). The difference

is mainly caused by the amount of MYC protein loaded and the exposure time for the blots. As shown in the above picture, the upper bands of c-Myc become more apparent with the increasing volumes of the loaded sample. To maintain data consistency, we have now replaced the old blot with a new one (revised Fig 2H) in which the upper bands of c-Myc are more apparent. The new experiments were conducted in three independent biological replicates and quantified.

8. How is Myc expression in the shPRODH group compared to the control group (in both naïve and primed cells)?

Response: The Myc expression levels in the shPRODH group are consistently elevated compared to the control group in both naïve and primed states (revised Appendix Fig2B). This result indicates a new layer of correlation between MYC and PRODH, and the potential feedback regulatory mechanism requires further investigation. We thank the reviewer for this stimulating question and have added the information to the Discussion section of the manuscript.

9. When limiting ROS production by MitoQ treatment, are there any effects on mtOXPHOS and glycolysis (feedback loops)? This is not a critical question and it is up to the authors to decide whether they would like to address it experimentally.

Response: The interplay between ROS and central carbon metabolism is rather complicated. ROS inhibits essential glycolysis and oxidative phosphorylation kinases but activates the HIF1 pathway and stimulates cellular glucose uptake (Bhardwaj & He, 2020; Liemburg-Apers *et al*, 2015). In this study, given MitoQ's capacity to restore pluripotency and cellular proliferation (revised Fig 5F and 5G), it is likely to be able to partially restore oxidative phosphorylation and glycolysis levels. In the Discussion section, we speculated that PRODH might directly or indirectly influence oxidative post-translational modifications of ETC components (Lennicke & Cochemé, 2021), thereby regulating their protein stability. MitoQ may have the potential to alleviate this phenomenon and rectify the aberrant mtOXPHOS and glycolysis.

10. PRODH k.d. show no effect on proline metabolism. Why? Are there any compensatory factors? This can be discussed.

Response: Even if there are elevated level of intracellular proline as the result of PRODH knock-down, multiple metabolism pathways can bring its level down to maintain homeostasis. As shown in the above KEGG pathway, apart from being metabolized to P5C by PRODH, proline can undergo hydroxylation modification, forming hydroxyproline. This hydroxyproline can subsequently be metabolized into 4-hydroxy-2-oxoglutarate. Furthermore, 4-hydroxy-2-oxoglutarate can undergo breakdown into pyruvate and glyoxylate for further metabolism.

Redacted: a snippet from the KEGG: Arginine and proline metabolism pathway

11. Can the authors provide any statistics on EV6D (ROS levels)?

Response: As shown in revised Fig EV5C, Rset-shPRODH and PXGL-shPRODH cells were treated with 50 nM MitoQ for 3 hours and cultured for an additional 12 hours. The fluorescent intensity of cells stained with MitoSOX was determined by flow cytometry (upper), and quantitative data normalized to Control cell values (H9N Rset-PLKO.1 or H9N PXGL-PLKO.1) were presented (lower). The experiments were conducted in three independent biological replicates and quantified, and the raw data were uploaded.

Reviewer #3: The manuscript from Chen and colleagues describes a study to investigate the role of PRODH - a mitochondrial enzyme that oxidises proline - in embryonic stem cell phenotype. While it has previously been shown that PRODH expression is regulated by MYC, it has not been described in ESCs, or through a culture supplement (the MEK inhibitor, PD0325901).

Comments:

While this is a potentially exciting story, the issues of data quality throughout mean that it is difficult to be convinced that the narrative is correct. There are some exciting experiments in there that really suggest something is going on, but overall the data quality and experimental design needs to be improved. Particular concern needs to be taken over the mass spectrometry metabolomic data, which is problematic (as outlined below).

Response: We appreciate the reviewer's valuable comments and have tried our best to address these shortcomings. To further consolidate our data and the associated conclusions, we re-designed and conducted new experiments following the reviewer's suggestions. Besides, we performed rigorous quantitative analysis and statistical significance test on almost all experimental data and incorporated them into the revised manuscript, as presented in revised Fig 1B/E/F/H, Fig 2B-J, Fig 3C/F/G, Fig 5A/C/E-G, Fig 6, Fig EV1B-F, Fig EV3, Fig EV4, Fig EV5A/C, Appendix Fig2, and Appendix Fig3. We specified biological or technical replicate numbers and the *P* values of statistical analyses in the above-mentioned figures and their legends. In addition, we also uploaded all source data (including the replicate data). We believe that such efforts made our data more robust and conclusions more convincing.

Overall the style of the manuscript makes it impenetrable for a non-expert of the ESC field, particularly at the beginning. The language needs to be decoded throughout to make it understandable for more than a niche audience.

Response: Thanks for the suggestion. We have carefully proofread the manuscript and made multiple corrections and modifications to make it more readable.

The authors note that knockdown of PRODH decreased AP+ colony formation (Figure 1E). Did it reduce colony number overall? It appears to have reduced colony size from the image. They also in the same section note that it reduced NANOG expression in the PXGL-cultured ESCs - did it also change colony size and AP+ expression, given that these are both models of naïve-like ESCs? What about NANOG expression in the Rset culture? Uncertain, given evidence in Ex Figure 1A whether this effect is only in a single model (PXGL) as the loading is uneven in the Rset blot, and the reduction in NANOG may not be significant. Indeed in H9P primed cells, all three markers may increase in expression. The images shown in Ex Figure 1E are highly unclear, alongside that of Figure 1E - these need to be improved to be informative.

Response: We are grateful for the reviewer's constructive comments. AP staining, immunofluorescence microscopy, and Western blotting were re-designed and conducted to validate alterations in the pluripotent characteristics of hESCs.

For AP staining, we replaced the dye, conducted a new set of experiment, and quantified the results. ***The blue dye highlighted a pronounced reduction in both size and number of AP+ colonies in the naïve state following PRODH knock-down*** (revised Figs 1E and EV1A). For immunofluorescence microscopy, we repeated the experiments using hESCs treated and cultured under six conditions. ***Notably, between naïve hESC clumps with comparable size and morphology, shPRODH-transfected cells had significantly diminished NANOG immunofluorescence signals compared to PLKO.1-transfected control cells*** (revised Figs 1F). Additionally, we repeated Western blot experiments, comparing hESCs treated and cultured under six conditions on the same gel and quantified the results. Tubulin was served as a sample loading control to normalize the differences in total cell/protein numbers of different samples loaded onto the gel. ***The levels of NANOG and SOX2 in Rset/PXGL-shPRODH cells were approximately 70~80% of those of control group*** (revised Figs EV1B and EV1C). All experiments above were conducted in three independent biological replicates, and the raw data have been uploaded.

Taken together, our comprehensive analyses led us to conclude that PRODH knock-down impairs human naïve pluripotency in both Rset and PXGL culture media.

Reproducibility of the data are unclear throughout - it is not apparent how many times any experiment was performed except in a minority of cases, whether replicates were taken within each experiment, etc.

Response: Our data presented in the entire manuscript included a minimum of three replicates, either biological or technical replicate. More detailed information about the reproducibility of data has been added to the Data information in the Figure legend section. Besides, in most bar graphs showing the mean values and the S.D. bars, we also presented the triplicate values as individual data points, facilitating visual assessment of the data variation and reproducibility.

In describing their metabolomic data (EV2A and B) the authors suggest that as there are more upregulated than downregulated metabolites, there is 'more active metabolic activity' within naïve hESCs. This is not an accurate interpretation of these data - decreases in metabolites can also indicate pathway activity (due to a higher rate of use) and increased metabolites can indicate blocked pathways. They should reconsider this description.

Response: We are most grateful to the reviewer for pointing out our incorrect interpretation of the metabolomics data. Indeed, measurement of metabolite concentrations by metabolomics tells only half the story. Equally important is understanding pathway activity, which can be quantified in terms of material flow per unit time, i.e. metabolic flux. Concentrations and fluxes do not reliably align. For example, a common cause of metabolite build-up is decreased consumption. Because metabolite levels and fluxes provide complementary information, metabolic understanding is best achieved by investigating both. We made corresponding corrections in the text.

The list of metabolites appears to be via putative ID, and therefore there are a large number of metabolites that should not be possible to observe in mammalian cell culture, including ibulast (a drug), bakers yeast extract, vinyl acetic acid, clidinium (another drug), etc. Further examples of this lie in the EV4E plot, showing compounds such as rhodamine b, suprofen and penicillin g varying between the two groups. It significantly reduces confidence in the expertise used to analyse the dataset, and the identity of any of the metabolites observed and reported. Given this, and the reported difficulty in detecting P5C in cell extracts in the field, the result shown in Figure 3A (P5C in H9N cells) needs to be confirmed with a chemical standard.

Response: We are so grateful for the careful examination kindly offered by the reviewer.

In our study, we employed two metabolomics methods: Targeted metabolomics to analyze amino acids and their derivatives in naïve and primed H9 cells, and untargeted metabolomics to identify changes in metabolites in H9P/H9N, H9P PLKO.1/H9P shPRODH, and H9N PLKO.1/H9N shPRODH cells.

For data shown in Fig 3A and EV3 (now in revised Fig 3C and EV2), the absolute quantity of metabolites in the samples was determined by targeted metabolomics, which required analytical standards (including the chemical standard of P5C) (revised Dataset EVI). This

approach ensured the precision and credibility of the results.

Untargeted metabolomics was conducted using ACQUITY UPLC T3 chromatography and Q Exactive mass spectrometer (Thermo Fisher Scientific, USA). For metabolite identification, Compound Discoverer 3.1.0 software (Thermo Fisher Scientific, USA) was employed. The process included importing raw data (.raw file) from LC-MS/MS mass spectrometry acquisition, followed by sequential steps: peak extraction and filtration, sample retention time alignment, addition of ions, filling missing values, and metabolite identification. ***Metabolite identification utilized an array of resources, including an in-house metabolite secondary mass spectrometry library, mzCloud, mzVault, Mass list, ChemSpider (HMDB, KEGG, LipidMaps) databases, and the mzLogic algorithm.*** This comprehensive approach yielded molecular weight, retention time, peak area, and metabolite identification for individual ions. The software parameters adhered to the default standards of Compound Discoverer 3.1.0. ***As metabolite databases do not distinguish between species, detecting metabolites that do not belong to mammals is possible. It is our oversight not to filter and exclude the matched metabolites that are of other species. Now, based on KEGG and manual selection, we have refined the dataset by excluding all metabolites that are of non-human origins and that are unlikely to be associated with human embryonic developmental stage.***

Due to the reduced number of the remaining metabolites (113), which were no longer suitable for macroscopic analyses between naïve and primed states, we cancelled the non-targeted metabolomics comparison between H9P and H9N. Given previous studies showed that PRODH knock-down had some effect on cellular metabolism, we focused on comparing the metabolite changes before and after PRODH knock-down in primed and naïve hESCs (revised Dataset EV2 and 3, Fig 3E, Fig 4A and B, and Fig EV3). Corresponding changes have been made in the revised manuscript.

Figure 3E shows that proline does not change between the comparisons (control versus shPRODH) and also does not appear to figure in the heatmap in EV data of those metabolites that changed. This strongly suggests that proline metabolism is not involved in the changes observed with shPRODH, thereby undermining the manuscript's conclusions. It therefore makes no logical sense to pursue this further as a line of investigation.

Response: The canonical role of PRODH in proline metabolism has been well established, and therefore our initial thought was proline metabolism might be an important regulator for human naïve pluripotency. However, this hypothesis was disproved by our metabolomics data. So, we consider it crucial to design an additional experiment for further validation (revised Fig3F and G).

After that, all the aforementioned results led us to investigate the most prominent phenotypes of PRODH knock-down that ultimately led us to reveal a non-canonical role of PRODH, namely, in limiting mtOXPHOS and ROS production, and thereby safeguarding naïve pluripotency of hESCs.

The enrichment of citric acid cycle metabolites shown (Figure 4A), which reflect a representation of the heatmap (EV4) also does not appear to link - other than succinate, there is no other change in TCA cycle metabolites that is present in the heatmap, and

it cannot be that one can infer a change in a whole metabolic pathway based on the changes in one metabolite that can be made through a number of other routes (e.g. 2-oxoglutarate-dependent dioxygenase activity).

Response: We agree with the reviewer's point. We therefore reanalyzed the data from untargeted metabolomics. Unlike the criteria established in the previous analysis for differential metabolites (P -value < 0.01 , $FC > 2$ or < 0.5), *we have now adjusted the filtration threshold to " P -value < 0.05 , $FC > 1.5$ or < 0.67 ". In contrast to only slight changes in metabolites (revised Figs EV3A-D), we found the most enriched pathway in naïve cells after PRODH knock-down was still the citric acid cycle, consistent with our previous conclusion, while it was changed to pantothenate and CoA biosynthesis in the primed state (revised Fig 4A). Four metabolites (fumaric acid, malic acid, succinic acid, and cis-aconitic acid), constituting 20% of the TCA pathway, exhibited significant upregulation after PRODH knock-down in the naïve state (revised Figs 4A and B, revised Figs EV3 F). Corresponding changes have been made in the revised manuscript.*

That being said, there is clearly a significant change in respiration in the shPRODH H9N cells (Rset condition) - more than doubling of the rate. Given that glycolysis shows around 20% decrease, these analyses suggest strongly that it is something other than ATP generation that is driving this change - therefore the authors conclusions of this section (pdf page 7) are not supported. The energy generation change by 2x OXPHOS is around 36X ATP if the coupling of the respiratory chain doesn't change. This is clearly not balanced by a 20% reduction in ATP generation by glycolysis. While not as extreme, a similar change is seen for the H9N PXGL state.

Response: We thank the reviewer for raising this point, and we realized that our initial description "This observation could be attributed to an intracellular metabolic reprogramming mechanism, wherein the levels of glycolysis are downregulated to counterbalance the impact of elevated oxidative phosphorylation, thus leading to the establishment of a new metabolic equilibrium." was misleading. For naïve mouse ESCs, it has been demonstrated that approximately 65% of cellular ATP is generated through oxidative phosphorylation, while the glycolytic pathway contributes around 35% of ATP (Cao *et al*, 2022). Therefore, 20% reduction in ATP generation by glycolysis will have limited effect on dramatically enhanced ATP production by OXPHOS resulting from PRODH knock-down. *In supporting this, our new data revealed that after knocking-down of shPRODH, both Rset- and PXGL-cultured hESCs exhibited about a 1.5-fold increase in total cellular ATP content compared to the control hESCs (revised Appendix Fig3).* We have removed the above initial description and added in the new result.

It is incorrect to say that 'mtOXPHOS [is] an energy metabolism that happens to be substantially weakened in the primed state'. Scientific terminology is important here, and referencing of statements such as these to support this type of remark (top of page 7 of the pdf).

Response: We appreciate the reviewer's point. We changed the original text to " *We speculated that PRODH might influence naïve pluripotency by regulating mtOXPHOS, an energy metabolism that proved to be significantly attenuated in the primed state (Tsogtbaatar et al,*

2020; Zhang et al, 2018; Zhou et al, 2012). This may explain why *PRODH* is specifically associated with naive pluripotency but has little impact on primed pluripotency." The relevant references were incorporated accordingly.

The ROS kit used to measure these species is based on DCFDA fluorescence, which mainly picks up cellular peroxide, and is not mitochondrial specific. The authors do not present data to support mitochondrial ROS production.

Response: We fully agree with the reviewer's pertinent and constructive comments. To improve this, we replaced the DCFDA with MitoSOX (superoxide indicators specifically targeted to mitochondria in live cells) and conducted additional experiments to determine the ROS levels. The new data that strongly support our previous conclusion were presented in the revised Figs 5A, 6G, and EV5C. All experiments above were conducted in three independent biological replicates, and the raw data were uploaded.

Co-localisation of mitochondria and lysosomes shown in Extended Figure 6C needs to be at higher resolution to be convincing - as the mitochondria cover most of the non-nuclear intracellular space, it is difficult to know for sure whether the colocalization is due to resolution rather than biology.

Response: We appreciate the reviewer's advice. To better visualize mitophagy, primed and naive (Rset and PXGL) hESCs were harvested for examination by transmission electron microscopy (TEM) (revised Figs 5E and EV5B). **Representative TEM images and quantitative analysis showed a striking increase in AVs and mitophagosome-like structures – AVs containing engulfed mitochondria after knocking down of *PRODH* in naïve hESCs.** Experimental details were included into the corresponding figure legends.

References

- Bhardwaj V, He J (2020) Reactive Oxygen Species, Metabolic Plasticity, and Drug Resistance in Cancer. *Int J Mol Sci* 21
- Bredenkamp N, Stirparo GG, Nichols J, Smith A, Guo G (2019) The Cell-Surface Marker Sushi Containing Domain 2 Facilitates Establishment of Human Naive Pluripotent Stem Cells. *Stem Cell Reports* 12: 1212-1222
- Cao J, Li M, Liu K, Shi X, Sui N, Yao Y, Wang X, Li S, Tian Y, Tan S et al (2022) Oxidative phosphorylation safeguards pluripotency via UDP-N-acetylglucosamine. *Protein & Cell*
- Farrell AS, Sears RC (2014) MYC degradation. *Cold Spring Harb Perspect Med* 4
- Gafni O, Weinberger L, Mansour AA, Manor YS, Chomsky E, Ben-Yosef D, Kalma Y, Viukov S, Maza I, Zviran A et al (2013) Derivation of novel human ground state naive pluripotent stem cells. *Nature* 504: 282-286
- Lennicke C, Cochemé HM (2021) Redox metabolism: ROS as specific molecular regulators of cell signaling and function. *Molecular Cell* 81: 3691-3707
- Liemburg-Apers DC, Willems PHGM, Koopman WJH, Grefte S (2015) Interactions between mitochondrial reactive oxygen species and cellular glucose metabolism. *Archives of Toxicology* 89: 1209-1226
- Liu W, Le A, Hancock C, Lane AN, Dang CV, Fan TW, Phang JM (2012) Reprogramming of proline and glutamine metabolism contributes to the proliferative and metabolic responses regulated by

oncogenic transcription factor c-MYC. *Proc Natl Acad Sci U S A* 109: 8983-8988

Ludwig TE, Levenstein ME, Jones JM, Berggren WT, Mitchen ER, Frane JL, Crandall LJ, Daigh CA, Conard KR, Piekarczyk MS *et al* (2006) Derivation of human embryonic stem cells in defined conditions. *Nat Biotechnol* 24: 185-187

Sladitschek HL, Neveu PA (2019) A gene regulatory network controls the balance between mesendoderm and ectoderm at pluripotency exit. *Mol Syst Biol* 15: e9043

Tsogtbaatar E, Landin C, Minter-Dykhouse K, Folmes CDL (2020) Energy Metabolism Regulates Stem Cell Pluripotency. *Front Cell Dev Biol* 8: 87

Zhang H, Menzies KJ, Auwerx J (2018) The role of mitochondria in stem cell fate and aging. *Development* 145

Zhou W, Choi M, Margineantu D, Margaretha L, Hesson J, Cavanaugh C, Blau CA, Horwitz MS, Hockenberg D, Ware C *et al* (2012) HIF1 α induced switch from bivalent to exclusively glycolytic metabolism during ESC-to-EpiSC/hESC transition. *The EMBO Journal* 31: 2103-2116

Dear Prof. Wang,

Thank you for submitting your revised manuscript. It has now been seen by all of the original referees.

My apologies for the delay in getting back to you - it took longer than anticipated to receive the referee reports given this busy time of the year.

As you can see, the referees find that the study is significantly improved during revision and recommend publication. However, I need you to address the points below before I can accept the manuscript.

- Please address the remaining concerns of referees #3 as below and provide a point-by-point response. Also, please highlight the changes in the text.
- Comment 1: Please provide higher quality images for Figure EV1 and also provide quantification.
- Comment 2: Please address this concern by making the changes in the text as requested by referee #3.
- Comment 3: Please address this concern by rephrasing the text and adding a discussion point about the caveat.
- Comment 4: Please include the information requested by referee #3 in the Materials & Methods.
- Comment 5: Please discuss this caveat in the text.
- Comments 6: Please add a discussion point on the possible contribution of PRODH to the ETC activity.
- Comment 7: Please respond to this comment textually.

Please let me know if you would like to discuss any of the points further.

- We note that the co-corresponding author Prof. Liming Huang currently does not have an institutional email address in their profile in the manuscript submission system. It is EMBO Press policy that all corresponding authors should add an institutional email address to their profile.
- Similarly, EMBO Press policy asks for all corresponding authors to link to their ORCID iDs. We note that the co-corresponding author Prof. Liming Huang's ORCID iD is currently not linked. You can read about the change under "Authorship Guidelines" in the Guide to Authors here: <https://www.embopress.org/page/journal/14693178/authorguide#authorshipguidelines>

In order to link your ORCID iD to your account in our manuscript tracking system, please do the following:

1. Click the 'Modify Profile' link at the bottom of your homepage in our system.
2. On the next page you will see a box halfway down the page titled ORCID*. Below this box is red text reading 'To Register/Link to ORCID, click here'. Please follow that link: you will be taken to ORCID where you can log in to your account (or create an account if you don't have one)
3. You will then be asked to authorise Wiley to access your ORCID information. Once you have approved the linking, you will be brought back to our manuscript system.

We regret that we cannot do this linking on your behalf for security reasons.

- Please remove the Author Contributions section from the manuscript.
- Please fill out and include an author checklist as listed in our online guidelines (<https://www.embopress.org/page/journal/14693178/authorguide>)
- We note that the funding information is not complete in the manuscript submission system - the National Key Research and Development Program of China (grant numbers 2016YFA0100303; 2016YFA0101201), the Zhejiang Provincial Natural Science Foundation of China (LY23H160002), the Postdoctoral Program of Shaoxing People's Hospital (2022BSH001), and a grant from the Independent Task of the State Key Laboratory for Diagnosis and Treatment of Infectious Diseases, the First Affiliated Hospital, School of Medicine, Zhejiang University.
- We note that Datasets EV1, 2 and 3 are currently uploaded them as Expanded View content. Please resubmit them as Dataset EV1, Dataset EV2 and Dataset EV3.
- Please resubmit Source Data as one file per figure. Source data files need to be submitted as zipped folders, one .zip file for each figure. Inside each folder, the files should be organized in subfolders, one subfolder for each panel.
- Supporting Information section needs to be removed from the manuscript.
- Please add zoom highlight boxes to Figure EV5B.
- Please make the datasets MTBLS7840 and MTBLS7832 publicly available and remove the login information from the manuscript.
- Our production/data editors have asked you to clarify several points in the figure legends:
 - o Please indicate the statistical test used for data analysis in the legends of figures EV 2; EV 3c-d.
 - o Please note that in figures 4f-h; EV 4d-f, h; there is a mismatch between the annotated p values in the figure legend and the annotated p values in the figure file that should be corrected.
 - o Please note that the box plots need to be defined in terms of minima, maxima, centre, bounds of box and whiskers, and percentile in the legends of figures 3e; EV 2.
 - o Please note that information related to n is missing in the legends of figures 4c-e; EV 2.
 - o Please note that scale bar and its definition are missing for figure EV 1a.

- Papers published in EMBO Reports include a 'synopsis' and 'bullet points' to further enhance discoverability. Both are displayed on the html version of the paper and are freely accessible to all readers. The synopsis includes a short standfirst summarizing the study in 1 or 2 sentences (max 35 words) that summarize the paper and are provided by the authors and streamlined by the handling editor. I would therefore ask you to include your synopsis blurb and 3-5 bullet points listing the key experimental findings.
- In addition, please provide an image for the synopsis. This image should provide a rapid overview of the question addressed in the study but still needs to be kept fairly modest since the image size cannot exceed 550 (width) x 300-600 (height) pixels.

Thank you again for giving us to consider your manuscript for EMBO Reports, I look forward to your minor revision.

Kind regards,

Deniz Senyilmaz Tiebe

--

Deniz Senyilmaz Tiebe, PhD
Editor
EMBO Reports

Referee #1:

The authors have addressed the points raised.

Referee #2:

The Authors addressed all of my questions.

Keep up the good work!

Referee #3:

1. Unlabelled figures make it very difficult to appropriately review this manuscript. Figure EV1A is not helpful - size and quality meant that it is can't be used to support conclusions.
2. Like all metabolites, proline is not 'expressed' like a gene, its concentration is changed within cells through synthesis, uptake or degradation (top of page 6 of new manuscript). Please could the authors go through the manuscript and alter instances where expression, upregulation and downregulation are used and change them for terms that indicate increases or decreases in amounts, levels or concentrations.
3. The authors cannot make the inference that proline is being differentially metabolised from the data shown. While many of the metabolites within the proline subnetwork change, indicating changes in this network as a whole, this evidence is insufficient to provide information on pathway activity, as agreed by the authors in their rebuttal to a previous comment. To discuss fluxes based on data in Figure 3C, authors must use appropriate methodology - such as modelling of changes in extracellular metabolites and/or stable isotope methods (page 6)
4. Regarding confirmation of metabolite ID in their metabolomic data, the authors say in their rebuttal that a chemical standard of P5C was used to confirm its ID (along with the other metabolites). The materials and methods section refers to the use of KEGG and HMDB, but not the use of chemical standards. Given that P5C is not frequently commercially available due to its instability, it is not clear where this chemical standard came from. It is therefore important for the authors to include this information in the materials and methods section.
5. The results shown in Figure 3G cannot be compared to the effect of loss of PRODH. Increasing the concentrations of a substrate exogenously does not necessarily mean that there is a change in the activity of one of the associated enzymes inside the cell. To conclude this would need a knockdown of PRODH in the presence (or absence) of exogenous proline. In such a way (and given the other metabolomic data), the authors have not ruled out the lack of a canonical role for PRODH with their data as they don't directly measure its activity.
6. Data shown in Figure 3C suggest that there are changes in the metabolic network around proline in between naïve and primed states, while the shPRODH investigation shows that there are wider metabolic changes (Figure EV3C-D). Through the oxidation of proline, PRODH activity is thought to pass electrons to the electron transport chain, which would be expected (through its canonical activity) to alter mitochondrial ROS and/or ATP generation (as previously highlighted in other publications), which could have effects on the wider metabolic network. It remains difficult to bring the metabolomic results alongside that of respiration etc (Figure 4) also shown.
7. shPRODH leads to increased respiration, particularly in the Rset cells, which occurs alongside an increase in TCA cycle metabolites. As increased respiration suggests increased oxidation of NADH produced by the TCA cycle, which is often

associated with increased flux through the TCA cycle. It is unclear how the authors link these two results. This increase in OXPHOS generated ATP (shown in Appendix Figure S3) is clearly an important output of the increased electron flux through the ETC, which is occurring after the knockdown of PRODH, so it is important that the data are all consistent with this.

Point-by-point response to reviewer #3

Reviewer #3 comment 1: Unlabelled figures make it very difficult to appropriately review this manuscript. Figure EV1A is not helpful - size and quality meant that it is can't be used to support conclusions.

Editor suggestion 1: Please provide higher quality images for Figure EV1 and also provide quantification.

Response: We presented higher magnification images of Fig EV1A to better illustrate the differences between the experimental and control groups. The corresponding quantitative data is presented in Fig 1E.

Reviewer #3 comment 2: Like all metabolites, proline is not 'expressed' like a gene, its concentration is changed within cells through synthesis, uptake or degradation (top of page 6 of new manuscript). Please could the authors go through the manuscript and alter instances where expression, upregulation and downregulation are used and change them for terms that indicate increases or decreases in amounts, levels or concentrations.

Editor suggestion 2: Please address this concern by making the changes in the text as requested by referee #3.

Response: Thanks for this helpful suggestion. We have made corresponding changes in the entire text to distinguish metabolites from gene transcripts/proteins.

Reviewer #3 comment 3: The authors cannot make the inference that proline is being differentially metabolised from the data shown. While many of the metabolites within the proline subnetwork change, indicating changes in this network as a whole, this evidence is insufficient to provide information on pathway activity, as agreed by the authors in their rebuttal to a previous comment. To discuss fluxes based on data in Figure 3C, authors must use appropriate methodology - such as modelling of changes in extracellular metabolites and/or stable isotope methods (page 6).

Editor suggestion 3: Please address this concern by rephrasing the text and adding a discussion point about the caveat.

Response: Thanks for the suggestion. We have reorganized the data of Fig 3C and 3D, and

revised the corresponding text in the Results (p5-p6), emphasizing the significance of employing appropriate experimental methods for metabolic flux analysis in the Discussion section (p10).

Reviewer #3 comment 4: Regarding confirmation of metabolite ID in their metabolomic data, the authors say in their rebuttal that a chemical standard of P5C was used to confirm its ID (along with the other metabolites). The materials and methods section refers to the use of KEGG and HMDB, but not the use of chemical standards. Given that P5C is not frequently commercially available due to its instability, it is not clear where this chemical standard came from. It is therefore important for the authors to include this information in the materials and methods section.

Editor suggestion 4: Please include the information requested by referee #3 in the Materials & Methods.

Response: In the Materials and Methods section, we explicitly stated the data processing method for **targeted** metabolomics, specifying to "**Use the AA standards' correct retention time to identify the metabolites**". The utilization of KEGG and HMDB was mentioned during **untargeted** metabolomics data processing. For data shown in Fig 3C and EV2, the absolute quantity of metabolites in the samples was determined by **targeted** metabolomics, which required analytical standards (including the chemical standard of P5C) (revised Dataset EV1). Due to the instability of P5C, **we outsourced its chemical synthesis to Jiangsu Aikon Biopharmaceutical Research and Development Co., Ltd. The synthesized P5C was dissolved to form a 0.2 M 6N hydrochloric acid solution for subsequent analysis.** Information regarding all the amino acid and derivative standards utilized has been outlined in the methodology section of targeted metabolomics (p17-p18).

Reviewer #3 comment 5: The results shown in Figure 3G cannot be compared to the effect of loss of PRODH. Increasing the concentrations of a substrate exogenously does not necessarily mean that there is a change in the activity of one of the associated enzymes inside the cell. To conclude this would need a knockdown of PRODH in the presence (or absence) of exogenous proline. In such a way (and given the other metabolomic data), the authors have not ruled out the lack of a canonical role for PRODH with their data as they don't directly measure its activity.

Editor suggestion 5: Please discuss this caveat in the text.

Response: We agree with the reviewer's point, and changed the sentence of "After ruling out

the possibility that the impairment of naïve pluripotency was mediated by the canonical role of PRODH in proline metabolism” to “Given the little indication that the impairment of naïve pluripotency was mediated by the canonical role of PRODH in proline metabolism” (p6). We added in “Direct measuring, comparing and manipulating the enzymatic activity of PRODH in naïve hESCs versus primed hESCs may help to further clarify if and to what extent the canonical role of PRODH contributes to maintaining human naïve pluripotency.” to the Discussion section (p11).

Reviewer #3 comment 6: Data shown in Figure 3C suggest that there are changes in the metabolic network around proline in between naïve and primed states, while the shPRODH investigation shows that there are wider metabolic changes (Figure EV3C-D). Through the oxidation of proline, PRODH activity is thought to pass electrons to the electron transport chain, which would be expected (through its canonical activity) to alter mitochondrial ROS and/or ATP generation (as previously highlighted in other publications), which could have effects on the wider metabolic network. It remains difficult to bring the metabolomic results alongside that of respiration etc (Figure 4) also shown.

Editor suggestion 6: Please add a discussion point on the possible contribution of PRODH to the ETC activity.

Response: As the reviewer pointed out, the canonical PRODH activity is expected to be associated with mitochondrial ROS and ATP production. In fact, in the context of cancer cells, enhanced PRODH activity is associated with promoted electron transfer in the ETC and increased ROS levels. For instance, as we mentioned in the Discussion section, PRODH in a human osteosarcoma cell line (U2OS) induced cell senescence associated with the increase in ROS production and accumulation of DNA damage (Nagano et al., 2017). In contrast, our findings in hESC context showed opposite pattern: PRODH limits mtOXPHOS and ROS production, and thereby safeguarding naïve pluripotency of hESCs. This is why we proposed that the role of PRODH in maintaining human naïve pluripotency may differ from its canonical role in proline metabolism. **We found that PRODH KD increased the protein stability of the ETC components, thereby augmenting electron transfer (Fig 6).** A similar phenomenon was reported in colorectal cancer cells (Hancock *et al*, 2015). The possible contribution of PRODH to the ETC activity was speculated in the Discussion section (p10).

Reviewer #3 comment 7: shPRODH leads to increased respiration, particularly in the Rset cells, which occurs alongside an increase in TCA cycle metabolites. As increased respiration suggests increased oxidation of NADH produced by

the TCA cycle, which is often associated with increased flux through the TCA cycle. It is unclear how the authors link these two results. This increase in OXPHOS generated ATP (shown in Appendix Figure S3) is clearly an important output of the increased electron flux through the ETC, which is occurring after the knockdown of PRODH, so it is important that the data are all consistent with this.

Editor suggestion 7: Please respond to this comment textually.

Response: As previously pointed out by the reviewer, assessing the activity of metabolic pathways solely based on the levels of metabolites, or vice versa, is not adequate. This is because, in cells with enhanced OXPHOS, both the production and consumption of intermediate metabolites in the TCA cycle increase concurrently. We agree with this point. That said, numerous published studies have demonstrated an augmentation of TCA cycle intermediate metabolites alongside enhanced OXPHOS (Geiger *et al*, 2016; Kuang *et al*, 2021). Therefore, we believe that our data and interpretation are rational and inherently consistent.

References:

Geiger R, Rieckmann JC, Wolf T, Basso C, Feng Y, Fuhrer T, Kogadeeva M, Picotti P, Meissner F, Mann M *et al* (2016) L-Arginine Modulates T Cell Metabolism and Enhances Survival and Anti-tumor Activity. *Cell* 167: 829-842.e813

Hancock CN, Liu W, Alvord WG, Phang JM (2015) Co-regulation of mitochondrial respiration by proline dehydrogenase/oxidase and succinate. *Amino Acids* 48: 859-872

Kuang W, Zhang J, Lan Z, Deepak R, Liu C, Ma Z, Cheng L, Zhao X, Meng X, Wang W *et al* (2021) SLC22A14 is a mitochondrial riboflavin transporter required for sperm oxidative phosphorylation and male fertility. *Cell Rep* 35: 109025

Dear Prof. Wang,

Thank you for submitting your revised manuscript. I have now looked at everything and all is fine. Therefore, I am very pleased to accept your manuscript for publication in EMBO Reports.

Congratulations on a nice work!

Kind regards,

Deniz Senyilmaz Tiebe

--

Deniz Senyilmaz Tiebe, PhD

Editor

EMBO Reports

--
